# Nonlinear ICA using Volume-Preserving Transformations

**Xiaojiang Yang**[1,*] **Yi Wang**[1]**, Jiacheng Sun**[2]**, Xing Zhang**[2]**, Shifeng Zhang**[2]**, Zhenguo Li**[2] **& Junchi Yan**[1,†]

[1] Shanghai Jiao Tong University, [2] Huawei Noah's Ark Lab
`{yangxiaojiang, refraction334, yanjunchi}@sjtu.edu.cn`
`{sunjiacheng1, zhangxing85, zhangshifeng4, li.zhenguo}@huawei.com`

## Abstract

Nonlinear ICA is a fundamental problem in machine learning, aiming to identify the underlying independent components (sources) from data which is assumed to be a nonlinear function (mixing function) of these sources. Recent works prove that if the sources have some particular structures (e.g. temporal structure), they are theoretically identifiable even if the mixing function is arbitrary. However, in many cases such restrictions on the sources are difficult to satisfy or even verify, hence it inhibits the applicability of the proposed methods. Different from these works, we propose a general framework for nonlinear ICA, in which the mixing function is assumed to be a volume-preserving transformation, and meanwhile the conditions on the sources can be much looser. We provide an insightful proof of the identifiability of the proposed framework. We implement the framework by volume-preserving flow-based models, and verify our theory by experiments on artificial data and synthesized images. Moreover, results on real-world images indicate that our framework can disentangle interpretable features.

## 1 Introduction

Independent component analysis (ICA) is one of the most fundamental problems in machine learning. The earlier works concentrate on linear ICA (Comon, 1994), in which the observed data is assumed to be a linear and invertible transformation (called "mixing function") of several independent components (called "sources"), and the goal is to identify the independent components from data points. Recently, there are increasing interests in nonlinear ICA (Hyvärinen & Pajunen, 1999), in which the mixing function is generalized to be nonlinear. This nonlinear problem is crucial, as it is the theoretical foundation of many important tasks. For example, one task is disentanglement (Locatello et al., 2019; Sorrenson et al., 2020; Locatello et al., 2020), which aims at learning some explanatory factors of variation from data (Bengio et al., 2013) and hence facilitates the interpretability of representation learning as well as its downstream tasks (Peters et al., 2017; Lake et al., 2017). Another example is controlling the generation process in generative models using semantic factors (Karras et al., 2019; Abdal et al., 2021). These tasks essentially require their latent variables to be identifiable in highly nonlinear latent models. Otherwise the latent variables can be mixing functions of explanatory / semantic factors, hence are probably not explanatory or semantic.

The central problem in nonlinear ICA is the unidentifiability. Specifically, if the observed data points are independent and identically distributed (i.i.d. in short), i.e. there is no temporal or similar structure in the data, then the sources are not identifiable essentially (Hyvärinen & Pajunen, 1999). This motivates many works to involve some structures to data for identifiability guarantee.

To deal with the unidentifiability problem, most existing works of nonliear ICA involve structures to data by restricting the sources in some plausible ways. One popular solution is to assume the sources have some kinds of temporal structures (Hyvarinen & Morioka, 2016; 2017). These works almost have no restriction on the mixing functions, but limit the data in the form of time series. Recent

---

*This work was done when Xiaojiang Yang was a research intern in Huawei Noah's Ark Lab.

†Corresponding author is Junchi Yan. The SJTU authors are with MoE Key Lab of Artificial Intelligence, Shanghai Jiao Tong University.

works (Hyvarinen et al., 2019; Khemakhem et al., 2020) extend the applicability of nonlinear ICA by involving auxiliary variables like labels of classes, and assume that the priors of sources are members of exponential family. However, their works require many classes in a dataset and each class should not be isolated in data space (Sorrenson et al., 2020), which is difficult to be satisfied. To see this, note that many datasets (e.g. CIFAR-10 (Krizhevsky et al., 2009)) just have very few classes, and usually each class is isolated in data space.

Another direction of involving structures to data is to restrict the mixing function. One way is to assume that the mixing function is post-nonlinear (Taleb & Jutten, 1999), i.e. each data variable is a nonlinear function of a linear combination of the sources. This is obviously overly restrictive, and hence is impractical. Some works provide a few heuristic settings on learning algorithms to facilitate identifiability (Watters et al., 2019; Sorrenson et al., 2020), e.g. Sorrenson et al. (2020) empirically report that volume preservation of flow-based models promotes disentanglement.

In this work, we explore a balancing direction: involve some natural structures to the mixing function, and meanwhile loosen the restrictions on the sources. Specifically, motivated by the empirical evidences by (Sorrenson et al., 2020), we assume the mixing function to be volume-preserving (and a weaker condition is provided in Appendix B), and seek some natural restrictions on the sources for identifiability guarantee. With mild conditions, we establish novel and fundamental identifiability theorems, which mainly show that if there exist two distinct classes in the datasets, the true sources are identifiable. The proofs are essentially different to existing works, and provide some insights for nonlinear ICA: the main indeterminacy of a nonlinear ICA framework using volume-preserving mixing functions is the rotation of latent variables.

Based on our identifiability theorems, we implement the proposed nonlinear ICA framework using a volume-preserving flow-based model (Dinh et al., 2014), which requires merely two classes of data according to our theory. First, we empirically show that with the most essential conditions on sources (two distinct classes with non-zero overlap) and mixing functions, the sources can be well identified, which implies that there might exist some extensions of our theorems with weaker conditions. Then we compare our implementation with the state-of-the-art identifiable model iVAE (Khemakhem et al., 2020) on artificial and synthetic image datasets, and show that our framework remarkably better results in terms of mean correlation coefficient. More importantly, we empirically show that besides the given two classes, involving more classes cannot improve identifiability. Moreover, experiments on MNIST (LeCun et al., 1998) and CelebA (Liu et al., 2015) indicate that the implementation is able to disentangle interpretable features using only two classes of data, which demonstrates the applicability of our theory. **Our contributions are as follows:**

1) We present a new solution to the unidentifiability problem of nonlinear ICA by using the general volume-preserving transformation, as widely used in flow-based models. Specifically, our theoretical results lessen the requirement by existing works on the sources by reducing the number of required classes to 2, at the slight cost of adding some natural conditions on the mixing function. Accordingly, all the restrictions are moderate, which facilitates the applicability of nonlinear ICA.

2) We establish two novel identifiability theorems. Specifically, by using the friendly and general volume-preserving property, they guarantee the sources can be identified up to a point-wise non-linearity and a point-wise linearity, respectively. The proofs suggest that the main indeterminacy of a nonlinear ICA framework with some natural conditions on mixing functions is simply the rotation of latent variables. The hope is to lay the foundation for a new path to identifiability.

3) We conduct experiments on both synthetic and real-world data to verify our theory, and point out that as long as the most essential conditions on sources and mixing functions are satisfied, our framework can well identify the true sources. This means that our theory can be further extended. Moreover, our framework remarkably excels the state-of-the-art nonlinear ICA method, which indicates that our framework is more powerful to identify the true sources.

## 2 BACKGROUNDS

In this section, we provide some backgrounds on nonlinear ICA theory. We first give a general definition of nonlinear ICA, and then explain its central problem, namely unidentifiability. Based on this, we discuss some recent proposed solutions, including their advantages and disadvantages.

## 2.1 DEFINITION AND THE KEY PROBLEM OF NONLINEAR ICA

Consider an observed random vector $\mathbf{x} \in \mathbb{R}^d$, we assume it is generated by an invertible *nonlinear* transformation $\mathbf{f}$ (called mixing function) using $n$ *independent* latent variables $\mathbf{s} = (\mathbf{s}_1, \cdots, \mathbf{s}_n)$ (called independent components or sources) as

$$\mathbf{x} = \mathbf{f}(\mathbf{s}), \tag{1}$$

where $n \leq d$ (Khemakhem et al., 2020). In earlier ICA theory it is commonly assumed that $n = d$, i.e. the number of independent components should be the same as the number of observed variables. As it is commonly believed that $n \ll d$ for real world data (Cayton, 2005; Narayanan & Mitter, 2010; Rifai et al., 2011), the definition above is much more practical.

The goal of nonlinear ICA theory is to identify (or recover) the independent components $\mathbf{s}_i$ based on the observations of $\mathbf{x}$ using a estimating function $\mathbf{g}$. Given the data distribution $p(\mathbf{x})$, if there exists a transformation $\mathbf{g}$ such that for arbitrary $\mathbf{f}$, $\mathbf{g} \circ \mathbf{f}$ is a point-wise linear transformation, then the mixing model (Eq. 1) is said to be identifiable.

Unfortunately, if there are no more restrictions on the mixing model (Eq. 1), it is seriously unidentifiable (Hyvärinen & Pajunen, 1999; Locatello et al., 2019). Specifically, even if the components of the estimated latent variables $\mathbf{z} \equiv \mathbf{g} \circ \mathbf{f}(\mathbf{s})$ are independent, $\mathbf{z}_i$ can be the mixing of $\{\mathbf{s}_i\}_{i=1}^n$. In other words, independence of components does not guarantee identifiability of the mixing model. E.g. suppose the prior $p(\mathbf{s})$ is a factorial multivariate Gaussian, take a point-wise scaling on $\mathbf{s}$ such that $p(\mathbf{s})$ becomes an isotropic multivariate Gaussian, then take a rotation around the center, the independence of components is unchanged, but the obtained components are the mixing of the original sources. As arbitrary density functions can be transformed to a Gaussian (Locatello et al., 2019), this example shows that for any prior, the independence of components is insufficient for identifiability.

## 2.2 EXISTING SOLUTIONS FOR UNIDENTIFIABILITY PROBLEM

To obtain identifiability, it is necessary to restrict the mixing model. There are two possible ways to involve restrictions: 1) restrict the sources $\mathbf{s}$; 2) restrict the mixing function $\mathbf{f}$.

In the first direction, one idea is to involve temporal structure to the sources $\mathbf{s}$, i.e. assuming the sources are time series $\mathbf{s}(t)$ with time index $t$ (Harmeling et al., 2003; Sprekeler et al., 2014; Hyvarinen & Morioka, 2016; 2017). This solution guarantees identifiability of the mixing model with few restrictions on the mixing function, but is limited in the setting of time series. A more general way is to involve an auxiliary variable $\mathbf{u}$ and assume that conditioned on them, the prior of sources $\mathbf{s}$ is a factorial member of exponential family (Hyvarinen et al., 2019; Khemakhem et al., 2020):

$$p(\mathbf{s}|\mathbf{u}) = \prod_{i=1}^n \frac{Q_i(\mathbf{s}_i)}{Z_i(\mathbf{u})} \exp\left(\sum_{j=1}^k T_{i,j}(\mathbf{s}_i)\lambda_{i,j}(\mathbf{u})\right), \tag{2}$$

where $T_{i,j}$, $\lambda_{i,j}$, $Z_i$ are the so-called sufficient statistics, coefficients and normalizing constant, respectively. $Q_i$ is the so-called base measure, and is simply set to be $1$ in many cases. $k$ is the order of the distribution, and distributions with higher $k$ are more flexible. The auxiliary variable $\mathbf{u}$ is additionally observed, which can be time index, class label, etc. This solution seems general, but it requires $nk + 1$ distinct values of $\mathbf{u}$ to guarantee identifiability, which is impossible in most datasets. For example, if we assume that in MNIST (LeCun et al., 1998), the prior of $\mathbf{s}$ is a factorial multivarite Gaussian ($k = 2$), and attempt to identify the sources using the 10 labels of the dataset based on the theory above, then we can identify 4 sources at most. However, the intrinsic dimensions is obviously far more than 4 (Pope et al., 2021), and hence we cannot identify the true sources.

In the second direction, there are very few works to obtain identifiability by restricting the mixing function $\mathbf{f}$ in nonlinear ICA theory. One solution is to assume $\mathbf{f}$ to be post-nonlinear, i.e. $\{\mathbf{x}_j\}_{j=1}^d$ are nonlinear functions of the linear combination of sources, then the mixing model is identifiable (Taleb & Jutten, 1999). This condition is too restrictive (Hyvarinen et al., 2019) in practice.

## 3 THE NONLINEAR ICA FRAMEWORK

In this section, we propose our general framework for nonlinear ICA (see Fig. 1).

### 3.1 DEFINITION OF GENERATIVE MODEL

We assume that the sources $\mathbf{s}$ are independent conditioned on an auxiliary variable $\mathbf{u}$:

$$p(\mathbf{s}|\mathbf{u}) = \prod_{i=1}^{n} p_i(\mathbf{s}_i|\mathbf{u}), \tag{3}$$

where $p_i$ is the $i$-th marginal distribution of the joint distribution $p$. Note that so far we do not restrict the density function of each source $p_i(\mathbf{s}_i|\mathbf{u})$ to any particular functional form. Hence our restrictions on the prior are looser than existing works (Hyvarinen et al., 2019; Khemakhem et al., 2020).

Next we introduce some coherent and basic restrictions on the mixing function $\mathbf{f}$. First, we assume it is a homeomorphism [1], mapping sources $\mathbf{s}$ to a vector of observed data variables $\mathbf{x}$ on a $n$-dimensional Riemannian manifold $\mathcal{M}$ embedded in $\mathbb{R}^d$:

$$\mathbf{f} : \mathbb{R}^n \to \mathcal{M} \subset \mathbb{R}^d$$
$$\mathbf{s} \mapsto \mathbf{x}. \tag{4}$$

This restriction is also natural, as it is the real world data often lies on a low dimensional manifold in high dimensional space (Rifai et al., 2011; Pope et al., 2021). We further assume it is volume-preserving (Dinh et al., 2014), i.e. the volume of each infinitesimal area on $\mathbb{R}^n$ equals to that of the area on $\mathcal{M}$ obtained by this transformation. The volume-preserving property is:

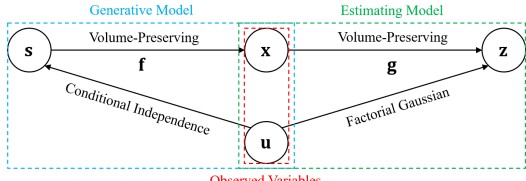

Figure 1: Structure and mild assumptions of our proposed framework. $\mathbf{x}$, $\mathbf{s}$, $\mathbf{u}$ and $\mathbf{z}$ are the observed data variables, the sources, the additionally observed variable and the estimated latent variables, respectively. $\mathbf{f}$ and $\mathbf{g}$ are the mixing function and estimating function, respectively, and both of them are volume-preserving. Conditioned on $\mathbf{u}$, the sources $\mathbf{s}$ are assumed to be independent, while the estimated latent variables $\mathbf{z}$ is required to follow a factorial multivariate Gaussian.

$$|\det \boldsymbol{J}_{\mathbf{f}}(\mathbf{s})| = 1, \tag{5}$$

where $J_{\mathbf{f}}$ is the Jacobian matrix of the mixing function $\mathbf{f}$ from the view of differential manifold[2].

Besides volume-preserving transformations, there exists a broader class of non-volume-preserving mixing functions to guarantee identifiability. We introduce such class of functions in Appendix B.

### 3.2 DEFINITION OF ESTIMATING MODEL

Given the generative model above, next we seek an estimating model to identify the sources $\mathbf{s}$ from $\mathbf{x}$. First, we set the estimating function $\mathbf{g}$ as a homeomorphism mapping the manifold $\mathcal{M}$ to $\mathbb{R}^n$:

$$\mathbf{g} : \mathcal{M} \to \mathbb{R}^n$$
$$\mathbf{x} \mapsto \mathbf{z}, \tag{6}$$

where $\mathbf{z}$ are estimated latent variables. Note the estimating function $\mathbf{g}$ is set to be volume-preserving:

$$\left|\det \boldsymbol{J}_{\mathbf{g}^{-1}}(\mathbf{z})\right| = 1. \tag{7}$$

Note that the volume-preservation here is a natural regularization to the estimating function for identifying the true sources. This is essentially different from the volume-preservation of mixing functions, which is a restriction on the generative process for identifiability.

To identify $\mathbf{s}$ by $\mathbf{z}$, naturally $\mathbf{z}$ should also be conditional independent. Besides, we set the distribution of $\mathbf{z}$ conditioned on $\mathbf{u}$ as a factorial multivariate Gaussian:

$$q(\mathbf{z}|\mathbf{u}) = \prod_{i=1}^{n} \frac{1}{Z_i(\mathbf{u})} \exp\left(-\frac{(\mathbf{z}_i - \mu_i(\mathbf{u}))^2}{2\sigma_i^2(\mathbf{u})}\right). \tag{8}$$

Actually, the model above gives a constraint to the priors $p(\mathbf{s}|\mathbf{u})$. Specifically, $q(\mathbf{z}|\mathbf{u})$ must be the push-forward of $p(\mathbf{s}|\mathbf{u})$ through $\mathbf{g} \circ \mathbf{f}$. As a result, the restriction on $\mathbf{z}$ above is actually an implicit

---

[1] A homeomorphism is a continuous function between topological spaces with a continuous inverse function.

[2] Here $\boldsymbol{J}_{\mathbf{f}}$ is a $n \times n$ Jacobian matrix of $\mathbf{f}$ relative to the charts (i.e. a neighborhood with a coordinate system) of $\mathbb{R}^n$ and $\mathcal{M}$, rather than a $d \times n$ Jacobian matrix relative to that of $\mathbb{R}^n$ and $\mathbb{R}^d$. So is $\boldsymbol{J}_{\mathbf{g}^{-1}}$.

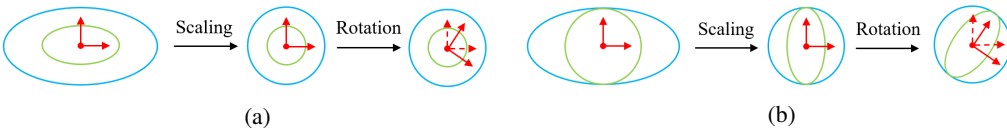

(a)                                             (b)

Figure 2: Sketch of the discussion about condition (iii). The blue and the green ellipses represent $q(\mathbf{z}|\boldsymbol{u}^{(1)})$ and $q(\mathbf{z}|\boldsymbol{u}^{(2)})$, respectively, and their lengths of major and minor axes correspond to $\sigma_1^{-1}$ and $\sigma_2^{-1}$. The red solid arrows represent the estimated latent variables $(z_1, z_2)$, while the red dashed arrows represent the obtained new latent variables $(z_1', z_2')$. The black solid arrows represent two operations: scaling to let one Gaussian (represented by blue ellipse) isotropic, and rotation around the center with arbitrary angle. **(a)** If condition (iii) is not satisfied, then the latent variables obtained by the two operations are still independent conditioned on both $\boldsymbol{u}^{(1)}$ and $\boldsymbol{u}^{(2)}$. **(b)** If condition (iii) is satisfied, then the obtained latent variables are no longer independent conditioned on $\boldsymbol{u}^{(2)}$.

one on $\mathbf{s}$. To see this, note that a volume-preserving transformation does not change the volume of a distribution's support. Hence, a multivariate uniform distribution cannot be transformed into a multivariate Gaussian by a volume-preserving function, because uniform distribution has limited support while the support of Gaussian is the whole space $\mathbb{R}^n$. Thus the priors $p(\mathbf{s}|\mathbf{u})$ cannot be a multivariate uniform distribution in our theory, and at least its support should cover $\mathbb{R}^n$.

## 4 THEORETICAL ANALYSIS

It is obvious that a generative model defined by Eq. 3-Eq. 5 is still unidentifiable using an estimating model defined by Eq. 6-Eq. 8. For example, if $\mathbf{f}$ is a point-wise nonlinear transformation mapping $p(\mathbf{s}|\mathbf{u})$ to an isotropic multivariate Gaussian, and $\mathbf{g}$ is a rotation around the center with arbitrary angle, then the conditions Eq. 3-Eq. 8 are satisfied, but the obtained $z_i$ is still the mixing of $\mathbf{s}$.

Therefore, we further seek some mild restrictions to guarantee the identifiability of the generative model. The goal is to prove that under some mild restrictions, the estimated latent variables $\mathbf{z}$ are the recoveries of the sources $\mathbf{s}$ up to a point-wise linear transformation, i.e. $\mathbf{g} \circ \mathbf{f}$ is point-wise linear function. Our main result is the following theorems:

**Theorem 1** *(**nonlinear identifiability**) Assume data points are sampled from a model defined by to Eq. 3-Eq. 8, and there exist two distinct observations of $\mathbf{u}$, denoted by $\boldsymbol{u}^{(1)}$ and $\boldsymbol{u}^{(2)}$, such that:*

   *(i) Both $\mathbf{f}$ and $\mathbf{g}$ have all second order derivatives.*

   *(ii) $\mu_i(\boldsymbol{u}^{(1)}) = \mu_i(\boldsymbol{u}^{(2)})$, $\forall\, i \in [n]$.*

   *(iii) $\left\{ \frac{\sigma_1(\boldsymbol{u}^{(2)})}{\sigma_1(\boldsymbol{u}^{(1)})}, \cdots, \frac{\sigma_n(\boldsymbol{u}^{(2)})}{\sigma_n(\boldsymbol{u}^{(1)})} \right\}$ are distinct.*

*Then $\mathbf{g} \circ \mathbf{f}$ is a composition of a point-wise nonlinear transformation and a permutation.*

**Remarks.** The theorem (proof is given in Appendix A) guarantees that the sources $\mathbf{s}$ of the generative model above are identifiable up to a nonlinear point-wise transformation using the estimating model above, and hence the generative model is identifiable. In the following, we discuss the meaning of the conditions (ii) and (iii) above, and omit condition (i) since it is very mild and trivial.

Condition (ii) means that the two multivariate Gaussians $q(\mathbf{z}|\boldsymbol{u}^{(1)})$ and $q(\mathbf{z}|\boldsymbol{u}^{(2)})$ are concentric, which guarantees that samples from the two Gaussians have large enough "overlap". Otherwise, if the centers of the two Gaussians have large distance, samples from these two Gaussians are likely to be absolutely disjoint when using Monte Carlo sampling. In this case, for the estimating function $\mathbf{g}$ there is equivalently one Gaussian because it can deal with samples from the two Gaussians separately, and hence identifiability is not guaranteed. In our proof (see Appendix A), however, it seems that this condition is not theoretically necessary, as the supports of the two Gaussians are the whole space and hence they theoretically have overlap everywhere. How to prove an identifiability theorem without this condition is very important, and we leave this problem to future works.

Condition (iii) means that there do not exist two estimated latent variables such that their variances of the two Gaussians are proportional. If this condition is not fulfilled, then we can easily construct another latent variables which are mixing of the estimated latent variables, and hence

the identifiability is impossibly guaranteed. As shown in Fig. 2, assume $\frac{\sigma_1(\boldsymbol{u}^{(2)})}{\sigma_1(\boldsymbol{u}^{(1)})} = \frac{\sigma_2(\boldsymbol{u}^{(2)})}{\sigma_2(\boldsymbol{u}^{(1)})}$, let $(z'_1, z'_2) = (z_1, z_2)\boldsymbol{SR}$, where $\boldsymbol{S} = \text{diag}\left(\frac{\sigma_1(\boldsymbol{u}^{(1)})^{-1}}{\sqrt{\sigma_1(\boldsymbol{u}^{(1)})^{-1}\sigma_2(\boldsymbol{u}^{(1)})^{-1}}}, \frac{\sigma_2(\boldsymbol{u}^{(1)})^{-1}}{\sqrt{\sigma_1(\boldsymbol{u}^{(1)})^{-1}\sigma_2(\boldsymbol{u}^{(1)})^{-1}}}\right)$ is a scaling matrix and $\boldsymbol{R}$ is an arbitrary rotation matrix of order 2, hence $\boldsymbol{SR}$ is a volume-preserving transformation. Then $(z'_1, z'_2)$ are independent conditioned on both $\boldsymbol{u}^{(1)}$ and $\boldsymbol{u}^{(2)}$, and hence they are also a set of estimated latent variables, but obviously they are the mixing of $(z_1, z_2)$. Therefore, condition (iii) is a natural and necessary restriction for identifiability.

Although (ii) and (iii) are conditions on the distributions of the estimated latent variables $\mathbf{z}$, they are also implicit restrictions on the prior of sources $\mathbf{s}$. To see this, note that $\mathbf{z} = \mathbf{g} \circ \mathbf{f}(\mathbf{s})$ and $\mathbf{g} \circ \mathbf{f}$ is a function constrained by Eq. 4-Eq. 6, and hence (ii) and (iii) implicitly restrict the prior through $\mathbf{g} \circ \mathbf{f}$.

If we further restrict the prior $p(\mathbf{s}|\mathbf{u})$ to a multivariate Gaussian, then we can reduce nonlinear indeterminacy to linear indeterminacy:

**Theorem 2** (*linear identifiability*) *Assume the hypotheses of Theorem 1 hold, and $p(\mathbf{s}|\boldsymbol{u}^{(1)})$ and $p(\mathbf{s}|\boldsymbol{u}^{(2)})$ are multivariate Gaussians. Then $\mathbf{g} \circ \mathbf{f}$ is a composition of a point-wise linear transformation and a permutation.*

In this case, the sources will have similar restrictions with the estimated latent variables. Specifically, let $\boldsymbol{\mu}^s(\mathbf{u})$ and $\boldsymbol{\sigma}^s(\mathbf{u})$ be the mean and variance of $p(\mathbf{s}|\mathbf{u})$, then we have: (ii') $\mu_i^s(\boldsymbol{u}^{(1)}) = \mu_i^s(\boldsymbol{u}^{(2)})$, $\forall i \in [n]$; and (iii') $\left\{\frac{\sigma_1^s(\boldsymbol{u}^{(2)})}{\sigma_1^s(\boldsymbol{u}^{(1)})}, \cdots, \frac{\sigma_n^s(\boldsymbol{u}^{(2)})}{\sigma_n^s(\boldsymbol{u}^{(1)})}\right\}$ are distinct.

The proofs of the theorems above (see Appendix A) give us some key insights for nonlinear ICA: *a nonlinear ICA framework using volume-preserving mixing functions only has three kinds of indeterminacy: point-wise nonlinear transformation, permutation and rotation of the latent variables.* Therefore, resolving the indeterminacy of rotation by condition (iii) will lead to identifiability, and the indeterminacy of point-wise nonlinear transformation can be reduced to point-wise linear transformation by the Gaussianity of $p(\mathbf{s}|\mathbf{u})$.

## 5 METHODOLOGY

As our theory requires an invertible estimating function, we set it as a volume-preserving Flow-based model (Dinh et al., 2014). The estimating function is denoted by $\mathbf{g}(\cdot; \boldsymbol{\theta})$, where $\boldsymbol{\theta}$ refers to model parameters. Given the data variables $\mathbf{x}$, $\mathbf{g}(\mathbf{x}; \boldsymbol{\theta})$ denotes the estimated latent variables $\mathbf{z}$.

Given a dataset $\mathcal{D} = \left\{(\boldsymbol{x}^{(1,1)}, \boldsymbol{u}^{(1)}), \cdots, (\boldsymbol{x}^{(1,M)}, \boldsymbol{u}^{(1)}), (\boldsymbol{x}^{(2,1)}, \boldsymbol{u}^{(2)}), \cdots, (\boldsymbol{x}^{(2,N)}, \boldsymbol{u}^{(2)})\right\}$ with only two labels $\boldsymbol{u}^{(1)}$ and $\boldsymbol{u}^{(2)}$, we can construct a loss by our theory. As in our theory, the conditional distribution of the estimated latent variables should be a factorial multivariate Gaussian, we have to push $\left\{\mathbf{g}(\boldsymbol{x}^{(1,i)}; \boldsymbol{\theta})\right\}_{i=1}^M$ and $\left\{\mathbf{g}(\boldsymbol{x}^{(2,i)}; \boldsymbol{\theta})\right\}_{i=1}^N$ to $q(\mathbf{z}|\boldsymbol{u}^{(1)})$ and $q(\mathbf{z}|\boldsymbol{u}^{(2)})$ (defined by Eq. 14) respectively. Thus, we minimize the negative log-likelihood following (Sorrenson et al., 2020):

$$\mathcal{L}(\boldsymbol{\theta}) = \mathbb{E}_{(\mathbf{x},\mathbf{u}) \in \mathcal{D}}\left[\sum_{i=1}^n \left(\frac{(g_i(\mathbf{x}; \boldsymbol{\theta}) - \mu_i(\mathbf{u}))^2}{2\sigma_i^2(\mathbf{u})} + \log(\sigma_i(\mathbf{u}))\right)\right] \tag{9}$$

To optimize the loss above, the number of sources $n$ is required, which is usually unknown. Fortunately, we can integrate dimension reduction into the loss, and hence the underlying dimensionality can be estimated by learning. Specifically, the loss above can be further simplified as follows, which has the ability of dimension reduction (see Appendix C for detailed derivation and discussion):

$$\mathcal{L}(\boldsymbol{\theta}) = \sum_{k=1}^2 \sum_{i=1}^n \log \sigma_i(\boldsymbol{u}^{(k)}; \boldsymbol{\theta}), \tag{10}$$

where $\sigma_i(\boldsymbol{u}^{(k)}; \boldsymbol{\theta}) = \sqrt{\mathbb{E}_{(\mathbf{x},\boldsymbol{u}^{(k)}) \in \mathcal{D}}\left[g_i(\mathbf{x}; \boldsymbol{\theta}) - \mathbb{E}_{(\mathbf{x},\boldsymbol{u}^{(k)}) \in \mathcal{D}}\left[g_i(\mathbf{x}; \boldsymbol{\theta})\right]\right]^2}$. We utilize the simplified version throughout our experiments.

## 6 EXPERIMENTS

Although our Theorem 1 does not explicitly restrict the functional form of $p(\mathbf{s}|\mathbf{u})$, and it implicitly requires $p(\mathbf{s}|\mathbf{u})$ to be identical to a factorial multivariate Gaussian, up to volume-preserving transfor-

mations. However, it is difficult to verify whether this implicit restriction is satisfied given a dataset. Therefore, our experiments are mainly based on Theorem 2, in which the prior is also assumed to be a factorial multivariate Gaussian. Hence, the main goal of this section is to show the performance of our framework to achieve linear identifiability in this case.

## 6.1 PROTOCOL

**Datasets.** We run experiments both on an artificial dataset and a synthetic image dataset called "Triangles", as well as on MNIST (LeCun et al., 1998) and CelebA (Liu et al., 2015). The generation processes of the artificial dataset and the synthetic images are described in Appendix D.

**Model specification.** To implement the volume-preserving estimating function $g(\cdot; \boldsymbol{\theta})$, we choose a volume-preserving flow called GIN (Sorrenson et al., 2020), which is a volume-preserving version of RealNVP (Dinh et al., 2017). For experiments on artificial data, the network is constructed by fully connected coupling blocks. For image datasets: Triangles and MNIST, the networks are constructed by both convolutional coupling blocks and fully connected coupling blocks. The parameters of each network are updated by minimizing the loss function in Eq. 10 using an Adam optimizer (Kingma & Ba, 2014). Details of the networks and their optimization refer to (Sorrenson et al., 2020).

**Performance metric.** To quantitatively compare the performance of different methods, we compute the mean correlation coefficient (MCC) (Khemakhem et al., 2020) between the sources and the estimated latent variables. The computation of MCC consists of three steps: i) calculate all pairs of correlation coefficients between the sources and estimated latent variables; ii) solve a linear sum assignment problem to assign one estimated latent variable to each source for highest total correlation coefficients; iii) take the average correlation coefficients of the obtained pairs. A high MCC means that we successfully identify the true sources, up to point-wise transformations.

## 6.2 RESULTS ON ARTIFICIAL DATASET AND TRIANGLES

On artificial dataset and Triangles, the labels of Gaussians in the prior of sources are used as the observations of the auxiliary variables $\mathbf{u}$ required in our theory. We mainly aim at verifying our theory, and comparing our framework with the state-of-the-art nonlinear ICA method, namely iVAE (Khemakhem et al., 2020), which is based on variational autoencoders (VAE) (Kingma & Welling, 2013).

### 6.2.1 ABLATION STUDY

To verify what conditions in our theory are essential for identifying the true sources, we perform an ablation study on artificial data. The chosen conditions include i) the concentric property of two Gaussians $p(\mathbf{s}|\boldsymbol{u}^{(1)})$ and $p(\mathbf{s}|\boldsymbol{u}^{(1)})$, denoted by $\mu_i^s(\boldsymbol{u}^{(1)}) = \mu_i^s(\boldsymbol{u}^{(2)})$; ii) the variances of the two Gaussains should be not proportional, denoted by $\frac{\sigma_1^s(\boldsymbol{u}^{(2)})}{\sigma_1^s(\boldsymbol{u}^{(1)})} \neq \frac{\sigma_2^s(\boldsymbol{u}^{(2)})}{\sigma_2^s(\boldsymbol{u}^{(1)})}$; iii) the the volume-preserving property of mixing function $\mathbf{f}$; iv) the existence of two distinct

Table 1: MCC score ↑ on artificial data when part of the conditions are not satisfied.

| Conditions | Mean±STD |
|---|---|
| Fully satisfied | 1.000±0.000 |
| $\mu_1^s(\boldsymbol{u}^{(1)}) \neq \mu_1^s(\boldsymbol{u}^{(2)})$ | 1.000±0.000 |
| $\frac{\sigma_1^s(\boldsymbol{u}^{(2)})}{\sigma_1^s(\boldsymbol{u}^{(1)})} = \frac{\sigma_2^s(\boldsymbol{u}^{(2)})}{\sigma_2^s(\boldsymbol{u}^{(1)})}$ | 1.000±0.000 |
| Non-volume-preserving $\mathbf{f}$ | 1.000±0.000 |
| $\boldsymbol{u}^{(1)} = \boldsymbol{u}^{(2)}$ | 0.999±0.001 |
| No overlap | 0.942± 0.064 |

observations of auxiliary variable $\boldsymbol{u}^{(1)}$ and $\boldsymbol{u}^{(2)}$; v) the existence of overlap of samples from $p(\mathbf{s}|\boldsymbol{u}^{(1)})$ and $p(\mathbf{s}|\boldsymbol{u}^{(1)})$. For verifying experiment of each condition, we synthesize an artificial dataset in which the condition is not satisfied.

The quantitative results of our framework when the conditions above are not satisfied are shown in Table 1, and the qualitative results are shown in Appendix E for the sake of brevity. Note that in the case of $\mu_1^s(\boldsymbol{u}^{(1)}) \neq \mu_1^s(\boldsymbol{u}^{(2)})$, samples from $p(\mathbf{s}|\boldsymbol{u}^{(1)})$ and $p(\mathbf{s}|\boldsymbol{u}^{(2)})$ have non-zero overlap.

Over these experimental results, we made the following observations:

1) As long as samples from $p(\mathbf{s}|\boldsymbol{u}^{(1)})$ and $p(\mathbf{s}|\boldsymbol{u}^{(2)})$ have non-zero overlap, GIN will almost perfectly identify the true sources, even if the $p(\mathbf{s}|\boldsymbol{u}^{(1)})$ and $p(\mathbf{s}|\boldsymbol{u}^{(2)})$ are not concentric. However, when the overlap vanishes, the performance of GIN has a sharp decline. These results indicate that the condition (ii) in our Theorem 1 is not necessary, but non-zero overlap is crucial for identifiability.

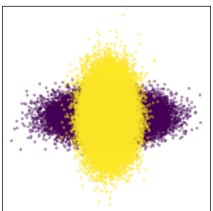 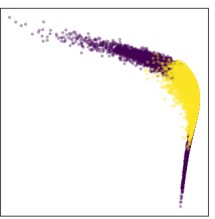 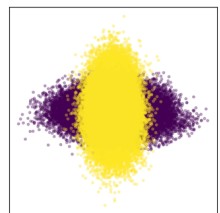 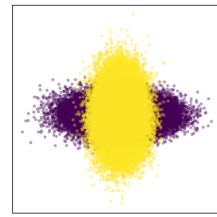

(a) Ground truth      (b) Data points     (c) Estimation by iVAE  (d) Estimation by GIN

Figure 3: Qualitative comparison of iVAE and GIN on artificial dataset with two classes. (a) ground truth (observations of the sources) consists of samples from two Gaussians, which are visualized by two different colors. (b) 2-dim projection of the 10-dim data points. (c) and (d) estimations of the ground truth by iVAE and GIN, respectively. The plotted estimated latent variables are chosen by assignment of correlation coefficients between sources and all estimated latent variables.

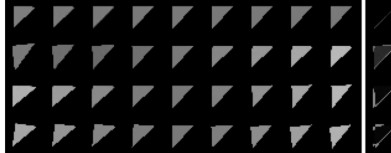 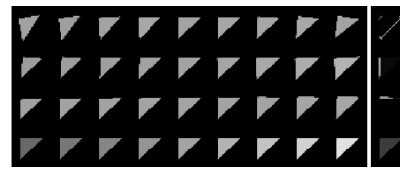

(a) Traversals by iVAE               (b) Traversals by GIN

Figure 4: Qualitative comparison of iVAE and GIN on Triangles images. Each row is traversal by manipulating one estimated latent variable. The last image of each row represents the heat map (the map of changed pixels) of that row, generating by taking the difference of the 3rd and 7th images. Obviously, the four rows correspond to rotation, width, height and gray level, respectively.

2) When $\frac{\sigma_1^s(\boldsymbol{u}^{(2)})}{\sigma_1^s(\boldsymbol{u}^{(1)})} \neq \frac{\sigma_2^s(\boldsymbol{u}^{(2)})}{\sigma_2^s(\boldsymbol{u}^{(1)})}$ or $\boldsymbol{u}^{(1)} = \boldsymbol{u}^{(2)}$, the sources are unidentifiable according to our theoretical analysis, but the experimental results show that in these cases the true sources still be well identified. This is not contradictory to the necessity of condition (iii) in Theorem 1 and two distinct classes, but indicates that there exist some biases in learning process to reduce the indeterminacy of rotation of latent variables. We conjecture that one bias is mini-batch sampling. As in mini-batch sampling, the empirical distributions of different mini-batches can be very different, while according to our theory, two different priors are almost sufficient to guarantee identifiability.

3) When the mixing function $\mathbf{f}$ (not the estimating function $\mathbf{g}$) is non-volume-preserving, the recovery of sources can be somewhat unsuccessful. To see this, note that in Fig. 7(d), the recovery of ground truth has some visual distortions. This demonstrates that involving some restrictions to $\mathbf{f}$ (like volume-preservation or the condition introduced in Appendix B) is necessary.

### 6.2.2 Qualitative results

**Results on artificial data.** As shown in Fig. 3, both iVAE and GIN can successfully identify the true sources from their nonlinear mixing data points. While GIN almost perfectly recovers the ground truth, yet the recovery by iVAE has some distortions everywhere.

**Results on Triangles.** For qualitative comparison of iVAE and GIN, we plot traversals of the estimated latent variables by them, which are obtained by three steps: i) deal with the assignment problem of correlation coefficients between 4 sources and all estimated latent variables, and choose the assigned latent variables for traversals, denoted by $\{z_i\}_{i=1}^4$; ii) estimate the variances of the chosen latent variables $\{\sigma_i\}_{i=1}^4$; iii) manipulate the value of the $i$-th latent variable between $(-2\sigma_i, 2\sigma_i)$ while keep other estimated latent variables unchanged, and then generate the $i$-th row of images.

In Fig. 4, the qualitative performances of iVAE and GIN are distinct. Each plot estimated latent variable of iVAE is a mixing of the true sources, e.g. the 4-th row of Fig. 4(a) is a mixing of rotation and color. Compared with this, the plot estimated latent variables by GIN obviously correspond to the true sources: rotation, width, height and gray level, respectively. This shows that our framework is much better than iVAE qualitatively, and is able to identify the true sources from images.

These results demonstrates that our framework is much more powerful than iVAE for identifying true sources on both artificial data and synthetic images with a few classes.

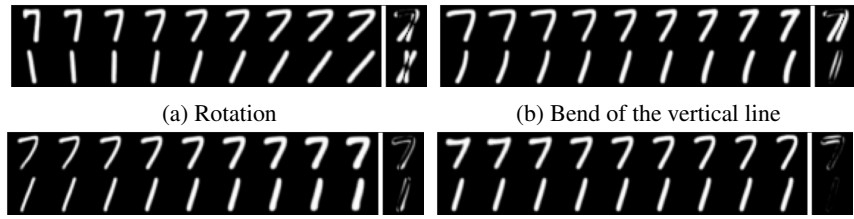

| (a) Rotation | (b) Bend of the vertical line |

| (c) Thickness | (d) Bend of the horizontal bar |

Figure 5: Traversals of estimated latent variables w/ top 4 standard deviations by GIN on MNIST.

### 6.2.3 QUANTITATIVE RESULTS

The quantitative results are shown in Table 2, in which '-$mc$' means the model is trained and tested on a dataset with $m$ classes. According to the theory by (Hyvarinen et al., 2019; Khemakhem et al., 2020), iVAE requires 9 classes on Triangles and 5 classes on our artificial data, hence we set iVAE-9c, iVAE-5c and iVAE-2c for fair comparisions with GIN. As for GIN, we set GIN-9c, GIN-2c and GIN-1c to verify whether 2 classes predicted by our theory are sufficient and necessary.

As shown in Table 2, for identifying true sources, GIN excels iVAE w.r.t. MCC on both artificial data and synthetic images. This might be due to the volume-preserving essence of GIN, which provides a strong but natural restriction for estimating functions according to our theory. Also, besides the given two classes, increasing the number of classes cannot markedly improve the performance of GIN, while reducing number of classes to 1 leads to a sharp decline of MCC. This means for GIN, two classes of data are sufficient and necessary for identifiability. These results are consistent with our theory.

Table 2: Mean and STD of MCC score on synthetic dataset and Triangles of 5 trials.

| Methods | Artificial Data | Triangles |
|---------|-----------------|-----------|
| iVAE-9c | - | 0.664±0.014 |
| iVAE-5c | 0.985±0.001 | - |
| iVAE-2c | 0.983±0.001 | 0.659±0.055 |
| GIN-9c | 1.000±0.000 | 0.857±0.031 |
| GIN-2c | 1.000±0.000 | 0.863±0.011 |
| GIN-1c | 0.999±0.001 | 0.787±0.033 |

### 6.3 RESULTS ON REAL WORLD DATASETS

To verify whether our framework is able to identify sources from real world datasets, we conduct experiments on MNIST and CelebA, and the setting and results on CelebA are reported in Appendix F. On MNIST, we pick images with digits '1' and '7' because they probably have non-zero overlap, and use their labels as observations of the auxiliary variable. As there is no ground truth in MNIST, we cannot compute the MCC score, and hence the qualitative results are mainly reported. Moreover, the estimated latent variables with high standard deviations are chosen to plot, as such variables are probably more meaningful (Sorrenson et al., 2020). We hope the estimated latent variables will correspond to some interpretable attributes in the dataset, which are viewed as true sources in disentanglement literature (Bengio et al., 2013; Locatello et al., 2019).

As shown in Fig. 5, the estimated latent variables with top 4 standard deviations are highly interpretable, corresponding to rotation, bend, thickness and bend of the horizontal bar. These results are comparable with fully utilizing the 10 classes of MNIST (Sorrenson et al., 2020). Therefore, here two classes are probably sufficient for identifying true sources, which is consistent with our theory.

## 7 CONCLUSION

We have explored a new direction in nonlinear ICA: restrict the mixing function to be volume-preserving, and meanwhile relax the restrictions on the sources. With mild conditions, we establish two novel identifiability theorems, which guarantee the sources can be identified up to point-wise non-linearity and a point-wise linearity, respectively. The proofs give insights for nonlinear ICA: if there exists some natural conditions on mixing functions, the main indeterminacy is simply the rotation of latent variables. This provides new paths and new techniques for identifiability guarantee.

For the applicability of our theory, we show in experiments that even most of the conditions are not satisfied, the true sources can still be successfully identified. This indicates that there exists stronger identifiability theorem with less conditions, which is an appealing for further exploration. We also empirically show that a volume-preserving flow-based model using two classes significantly excels the state-of-the-art ICA method (iVAE), and is able to learn interpretable attributes from real world datasets. These results show the advantages and applicability of our proposed framework.

ACKNOWLEDGMENTS

This work was supported by China Key Research and Development Program (2020AAA0107600), and Shanghai Municipal Science and Technology Major Project (2021SHZDZX0102).

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

# A   PROOF OF OUR IDENTIFIABILITY THEOREM

We first prove some lemmas on Euclidean space, and then prove our theorems on Riemannian manifold using these lemmas.

## A.1   LEMMAS ON EUCLIDEAN SPACE

Let $\mathbf{s} \sim p(\mathbf{s}|\mathbf{u})$, $\mathbf{s} \in \mathbb{R}^n$, and there exists a homeomorphism as follows:

$$\mathbf{f} : \mathbb{R}^n \to \mathbb{R}^n$$
$$\mathbf{s} \mapsto \mathbf{z}. \tag{11}$$

Suppose $\mathbf{f}$ is volume-preserving:

$$|\det \boldsymbol{J_f}(\mathbf{s})| = 1. \tag{12}$$

This volume-preserving property directly leads to a key corollary: denote the density of $\mathbf{z}$ conditioned on $\mathbf{u}$ by $q(\mathbf{z}|\mathbf{u})$, then $p(\mathbf{s}|\mathbf{u}) = q\left(\mathbf{f}(\mathbf{s})|\mathbf{u}\right)$. To see this, note that the change of variable rule gives us $p(\mathbf{s}|\mathbf{u}) = q\left(\mathbf{f}(\mathbf{s})|\mathbf{u}\right)|\det \boldsymbol{J_f}(\mathbf{s})|$, and substituting Eq. 12 leads to the conclusion. This is the starting point of our proof.

Further suppose components of $\mathbf{s}$ are independent, and $q(\mathbf{z}|\mathbf{u})$ is a factorial multivariate Gaussian:

$$p(\mathbf{s}|\mathbf{u}) = \prod_{i=1}^{n} p_i\left(\mathbf{s}_i|\mathbf{u}\right), \tag{13}$$

$$q(\mathbf{z}|\mathbf{u}) = \prod_{i=1}^{n} \frac{1}{Z_i^q} \exp\left(-\theta_{i,1}^q(\mathbf{u})\mathbf{z}_i - \theta_{i,2}^q(\mathbf{u})\mathbf{z}_i^2\right). \tag{14}$$

**Lemma 1** *Consider the model defined by Eq. 11-Eq. 14, if $\mathbf{f}$ have all second order derivatives, then there exist $\{\theta_{i,1}(\mathbf{u}), \theta_{i,2}(\mathbf{u})\}_{i=1}^{n}$ such that*

$$\sum_{i=1}^{n}\left(\theta_{i,1}(\mathbf{u})\frac{\partial^2 f_i}{\partial \mathbf{s}_j \partial \mathbf{s}_k}(\boldsymbol{s}) + \theta_{i,2}(\mathbf{u})\frac{\partial f_i}{\partial \mathbf{s}_j}(\boldsymbol{s})\frac{\partial f_i}{\partial \mathbf{s}_k}(\boldsymbol{s})\right) = 0, \, \forall j \neq k, \, \forall \boldsymbol{s} \in \mathbb{R}^n. \tag{15}$$

**Proof** Note that $p(\mathbf{s}|\mathbf{u}) = q\left(\mathbf{f}(\mathbf{s})|\mathbf{u}\right)$, according to Eq. 13-Eq. 14, we have:

$$\prod_{i=1}^{n} p_i(\mathbf{s}_i|\mathbf{u}) = \prod_{i=1}^{n} \frac{1}{Z_i^q} \exp\left(-\theta_{i,1}^q(\mathbf{u})f_i(\mathbf{s}) - \theta_{i,2}^q(\mathbf{u})f_i(\mathbf{s})^2\right). \tag{16}$$

Take the logarithm, we have:

$$\sum_{i=1}^{n}\left(\log p_i(\mathbf{s}_i|\mathbf{u})\right) = -\sum_{i=1}^{n}\left(\theta_{i,1}^q(\mathbf{u})f_i(\mathbf{s}) + \theta_{i,2}^q(\mathbf{u})f_i(\mathbf{s})^2 + \log Z_i^q\right) \tag{17}$$

Because $\mathbf{f}$ have all second order derivatives, $f_i$ has Taylor expansion: $f_i(\boldsymbol{s}_0 + \Delta\mathbf{s}) = f_i(\boldsymbol{s}_0) + \nabla f_i(\boldsymbol{s}_0)^{\top}\Delta\mathbf{s} + \Delta\mathbf{s}^{\top}\boldsymbol{H}_{f_i}(\boldsymbol{s}_0)\Delta\mathbf{s} + \mathcal{O}(\Delta\mathbf{s}^{\top}\Delta\mathbf{s})$. Let $\mathbf{s} = \boldsymbol{s}_0 + \Delta\mathbf{s}$ in Eq. 17, and suppose $f_i(\boldsymbol{s}_0) = 0$ without loss of generality, then take the Taylor expansion of $f_i$, we have:

$$\sum_{i=1}^{n}\left(\log p_i(\mathbf{s}_i|\mathbf{u})\right)$$
$$= -\sum_{i=1}^{n}\left(\boldsymbol{\eta}_i^q\Delta\mathbf{s} + \theta_{i,1}^q(\mathbf{u})\Delta\mathbf{s}^{\top}\boldsymbol{H}_{f_i}(\boldsymbol{s}_0)\Delta\mathbf{s} + \theta_{i,2}^q(\mathbf{u})\left(\nabla f_i(\boldsymbol{s}_0)^{\top}\Delta\mathbf{s}\right)^2 + \mathcal{O}(\Delta\mathbf{s}^{\top}\Delta\mathbf{s}) + C_i^q\right), \tag{18}$$

where $\boldsymbol{\eta}_i^q = \theta_{i,1}^q(\mathbf{u})\nabla f_i(\boldsymbol{s}_0)^{\top}$, and $C_i^q$ is a constant.

Note that the left hand side of the equation above does not contain cross terms $\{\Delta\mathbf{s}_j\Delta\mathbf{s}_k\}_{j\neq k}$, while the right hand side does. Therefore, the coefficients of cross terms on the right hand side should be zeros:

$$\sum_{i=1}^{n}\left(\theta_{i,1}^q(\mathbf{u})\frac{\partial^2 f_i}{\partial \mathbf{s}_j \partial \mathbf{s}_k}(\boldsymbol{s}) + 2\theta_{i,2}^q(\mathbf{u})\frac{\partial f_i}{\partial \mathbf{s}_j}(\boldsymbol{s})\frac{\partial f_i}{\partial \mathbf{s}_k}(\boldsymbol{s})\right) = 0, \, \forall j \neq k, \, \forall \boldsymbol{s} \in \mathbb{R}^n. \tag{19}$$

Let $\theta_{i,1}(\mathbf{u}) \equiv \theta_{i,1}^q(\mathbf{u})$ and $\theta_{i,2}(\mathbf{u}) \equiv 2\theta_{i,2}^q(\mathbf{u})$, then the conclusion is obtained. $\qquad\square$

**Remark:** Denote the mean of $q_i$ by $\mu_i$, and the variance by $\sigma_i^2$, then $\theta_{i,1}^q = -\frac{\mu_i}{\sigma_i^2}$ and $\theta_{i,2}^q = \frac{1}{2\sigma_i^2}$. Without loss of generality, we can set $\mu_i = 0, \forall i$, and hence $\theta_{i,1} = 0, \forall i$. Therefore, Eq. 15 can be further simplified as

$$\sum_{i=1}^{n} \left( \theta_{i,2}(\mathbf{u}) \frac{\partial f_i}{\partial \mathrm{s}_j}(\boldsymbol{s}) \frac{\partial f_i}{\partial \mathrm{s}_k}(\boldsymbol{s}) \right) = 0, \forall j \neq k, \forall \boldsymbol{s} \in \mathbb{R}^n. \tag{20}$$

**Lemma 2** *Suppose all conditions in Lemma 1 are satisfied, and there exists two distinct values of* $\mathbf{u}$*, denoted by* $\boldsymbol{u}^{(1)}$ *and* $\boldsymbol{u}^{(2)}$*, such that the following conditions are also satisfied:*

*i)* $\mu_i(\boldsymbol{u}^{(1)}) = \mu_i(\boldsymbol{u}^{(2)}), \forall i \in [n]$;

*ii)* $\left\{ \frac{\sigma_1(\boldsymbol{u}^{(2)})}{\sigma_1(\boldsymbol{u}^{(1)})}, \cdots, \frac{\sigma_n(\boldsymbol{u}^{(2)})}{\sigma_n(\boldsymbol{u}^{(1)})} \right\}$ *are distinct.*

*Then* $\forall \boldsymbol{s} \in \mathbb{R}^n$*, the Jacobian* $J_{\mathbf{f}}(\boldsymbol{s}) = \left( \frac{\partial f_i}{\partial \mathrm{s}_j}(\boldsymbol{s}) \right)_{n \times n}$ *is a generalized permutation matrix.*

**Proof** Without loss of generality, let $\mu_i(\boldsymbol{u}^{(1)}) = 0, \forall i \in [n]$, according to Lemma 1 and condition i), we have

$$\begin{cases} \sum_{i=1}^{n} \left( \theta_{i,2}(\boldsymbol{u}^{(1)}) \frac{\partial f_i}{\partial \mathrm{s}_j}(\boldsymbol{s}) \frac{\partial f_i}{\partial \mathrm{s}_k}(\boldsymbol{s}) \right) = 0 \\ \sum_{i=1}^{n} \left( \theta_{i,2}(\boldsymbol{u}^{(2)}) \frac{\partial f_i}{\partial \mathrm{s}_j}(\boldsymbol{s}) \frac{\partial f_i}{\partial \mathrm{s}_k}(\boldsymbol{s}) \right) = 0 \end{cases}, \forall j \neq k. \tag{21}$$

Let $\boldsymbol{\Sigma}(\boldsymbol{u}, \boldsymbol{s}) \equiv \mathrm{diag}\left( \sum_{i=1}^{n} \theta_{i,2}(\boldsymbol{u}) \left( \frac{\partial f_i}{\partial \mathrm{s}_1}(\boldsymbol{s}) \right)^2, \cdots, \sum_{i=1}^{n} \theta_{i,2}(\boldsymbol{u}) \left( \frac{\partial f_i}{\partial \mathrm{s}_n}(\boldsymbol{s}) \right)^2 \right)$ and $\boldsymbol{\Lambda}(\boldsymbol{u}) \equiv \mathrm{diag}\left( \theta_{1,2}(\boldsymbol{u}), \cdots, \theta_{n,2}(\boldsymbol{u}) \right)$, then the equation above can be written as

$$\begin{cases} J_{\mathbf{f}}(\boldsymbol{s})^\top \boldsymbol{\Lambda}(\boldsymbol{u}^{(1)}) J_{\mathbf{f}}(\boldsymbol{s}) = \boldsymbol{\Sigma}(\boldsymbol{u}^{(1)}, \boldsymbol{s}) \\ J_{\mathbf{f}}(\boldsymbol{s})^\top \boldsymbol{\Lambda}(\boldsymbol{u}^{(2)}) J_{\mathbf{f}}(\boldsymbol{s}) = \boldsymbol{\Sigma}(\boldsymbol{u}^{(2)}, \boldsymbol{s}) \end{cases}. \tag{22}$$

Due to $\theta_{i,2}(\boldsymbol{u}) = \frac{1}{\sigma_i^2(\boldsymbol{u})} > 0$, $\boldsymbol{\Sigma}(\boldsymbol{u}, \boldsymbol{s})$ and $\boldsymbol{\Lambda}(\boldsymbol{u})$ are always positive definite, hence the equations above have square roots on both the two hand sides as follows:

$$\begin{cases} \boldsymbol{\Lambda}(\boldsymbol{u}^{(1)})^{1/2} J_{\mathbf{f}}(\boldsymbol{s}) = \boldsymbol{U}(\boldsymbol{u}^{(1)}) \boldsymbol{\Sigma}(\boldsymbol{u}^{(1)}, \boldsymbol{s})^{1/2} \\ \boldsymbol{\Lambda}(\boldsymbol{u}^{(2)})^{1/2} J_{\mathbf{f}}(\boldsymbol{s}) = \boldsymbol{U}(\boldsymbol{u}^{(2)}) \boldsymbol{\Sigma}(\boldsymbol{u}^{(2)}, \boldsymbol{s})^{1/2} \end{cases}, \tag{23}$$

where $\boldsymbol{U}(u)$ is orthogonal matrices, i.e. $\boldsymbol{U}(u)^\top \boldsymbol{U}(u) = \boldsymbol{I}$.

Therefore, we have

$$J_{\mathbf{f}}(\boldsymbol{s}) = \boldsymbol{\Lambda}(\boldsymbol{u}^{(1)})^{-1/2} \boldsymbol{U}(\boldsymbol{u}^{(1)}) \boldsymbol{\Sigma}(\boldsymbol{u}^{(1)}, \boldsymbol{s})^{1/2} = \boldsymbol{\Lambda}(\boldsymbol{u}^{(2)})^{-1/2} \boldsymbol{U}(\boldsymbol{u}^{(2)}) \boldsymbol{\Sigma}(\boldsymbol{u}^{(2)}, \boldsymbol{s})^{1/2}. \tag{24}$$

This leads to

$$\boldsymbol{U}(\boldsymbol{u}^{(2)})^{-1} \boldsymbol{\Lambda}(\boldsymbol{u}^{(2)})^{1/2} \boldsymbol{\Lambda}(\boldsymbol{u}^{(1)})^{-1/2} \boldsymbol{U}(\boldsymbol{u}^{(1)}) = \boldsymbol{\Sigma}(\boldsymbol{u}^{(2)}, \boldsymbol{s})^{1/2} \boldsymbol{\Sigma}(\boldsymbol{u}^{(1)}, \boldsymbol{s})^{-1/2}. \tag{25}$$

Both the two hand sides of the equation above can be viewed as the singular value decomposition (SVD) of $\boldsymbol{\Sigma}(\boldsymbol{u}^{(2)}, \boldsymbol{s})^{1/2} \boldsymbol{\Sigma}(\boldsymbol{u}^{(1)}, \boldsymbol{s})^{-1/2}$. Note that due to condition ii), the diagonal entries of $\boldsymbol{\Lambda}(\boldsymbol{u}^{(2)})^{1/2} \boldsymbol{\Lambda}(\boldsymbol{u}^{(1)})^{-1/2}$ are distinct. Based on this, according to the uniqueness of SVD (Kong et al., 2017), $\boldsymbol{U}(\boldsymbol{u}^{(1)})$ is the composition of a permutation matrix and a signature matrix. Substitute such decomposition of $\boldsymbol{U}(\boldsymbol{u}^{(1)})$ into Eq. 24, we finally conclude that $J_{\mathbf{f}}(\boldsymbol{s})$ is a generalized permutation matrix. □

**Remarks:** Since $J_{\mathbf{f}}(\mathbf{s})$ is a generalized permutation matrix, in each row and each column it has exactly one nonzero entry. Without loss of generality, we can assume $\frac{\partial f_i}{\partial \mathrm{s}_i}(\boldsymbol{s}) \neq 0$, while $\frac{\partial f_i}{\partial \mathrm{s}_j}(\boldsymbol{s}) = 0, \forall j \neq i$. Therefore, $f_i(\mathbf{s})$ is exactly the function of $\mathrm{s}_i$ and is independent of $\{\mathrm{s}_j\}_{j \neq i}$, and hence it can be denoted by $f_i(\mathrm{s}_i)$. Therefore, under some mild conditions in Lemma 2, $\mathbf{z}$ is identical to $\mathbf{s}$ up to a nonlinear point-wise transformation (and a permutation).

A.2   THEOREMS ON RIEMANNIAN MANIFOLD

Suppose data points are distributed on a Riemannian manifold $\mathcal{M}$ which is homeomorphic to $\mathbb{R}^n$. The homeomorphic can be denoted by

$$\begin{aligned} \mathbf{f} : \mathbb{R}^n &\to \mathcal{M} \\ \mathbf{s} &\mapsto \mathbf{x}, \end{aligned} \tag{26}$$

which is further assumed to be volume-preserving:

$$|\det \boldsymbol{J_f}(\mathbf{s})| = 1. \tag{27}$$

The density of $\mathbf{s}$ conditioned on $\mathbf{u}$ is denoted by $p(\mathbf{s}|\mathbf{u})$, which is assumed to be factorial, i.e. $\{s_i\}_{i=1}^n$ are independent:

$$p(\mathbf{s}|\mathbf{u}) = \prod_{i=1}^{n} p_i(s_i|\mathbf{u}). \tag{28}$$

Suppose the data variable $\mathbf{x}$ can be encoded into a vector of latent variables $\mathbf{z}$ with another homeomorphism, denoted by

$$\begin{aligned} \mathbf{g} : \mathcal{M} &\to \mathbb{R}^n \\ \mathbf{x} &\mapsto \mathbf{z}, \end{aligned} \tag{29}$$

where the homeomorphism is also volume-preserving:

$$\left|\det \boldsymbol{J_{g^{-1}}}(\mathbf{z})\right| = 1, \tag{30}$$

and conditioned on $\mathbf{u}$, $\mathbf{z}$ follows a factorial multivariate Gaussian, denoted by

$$q(\mathbf{z}|\mathbf{u}) = \prod_{i=1}^{n} \frac{1}{Z_i} \exp\left(-\frac{z_i - \mu_i(\mathbf{u}))^2}{2\sigma_i^2(\mathbf{u})}\right). \tag{31}$$

**Theorem 1** (**Nonlinear Identifiability**) *Assume data points are sampled from a generative model defined according to Eq. 26-Eq. 31, and there exist two distinct observations of $\mathbf{u}$, denoted by $\boldsymbol{u}^{(1)}$ and $\boldsymbol{u}^{(2)}$, such that the following holds:*

*i) Both $\mathbf{f}$ and $\mathbf{g}$ have all second order derivatives.*

*ii) $\mu_i(\boldsymbol{u}^{(1)}) = \mu_i(\boldsymbol{u}^{(2)}), \forall i \in [n]$.*

*iii) $\left\{\frac{\sigma_1(\boldsymbol{u}^{(2)})}{\sigma_1(\boldsymbol{u}^{(1)})}, \cdots, \frac{\sigma_n(\boldsymbol{u}^{(2)})}{\sigma_n(\boldsymbol{u}^{(1)})}\right\}$ are distinct.*

*Then $\mathbf{g} \circ \mathbf{f}$ is a composition of a point-wise nonlinear transformation and a permutation.*

**Proof** Note that $\mathbf{g} \circ \mathbf{f} : \mathbf{s} \mapsto \mathbf{z}$ is a homeomorphism which is volume-preserving $|\det \boldsymbol{J_{g\circ f}}(\mathbf{s})| = 1$ and has all second order derivatives. According to Lemma 2, we can conclude that $\mathbf{g} \circ \mathbf{f}$ is a composition of a point-wise nonlinear transformation and a permutation. □

**Theorem 2** (**Linear Identifiability**) *Assume the hypotheses of Theorem 1 hold, and $p(\mathbf{s}|\boldsymbol{u}^{(1)})$ and $p(\mathbf{s}|\boldsymbol{u}^{(2)})$ are multivariate Gaussians. Then $\mathbf{g} \circ \mathbf{f}$ is a composite of a point-wise linear transformation and a permutation.*

**Proof** Let $\mathbf{h} = \mathbf{g} \circ \mathbf{f}$. According to Theorem 1, $\mathbf{h}$ is a composite of point-wise nonlinear transformation and a permutation, and hence without loss of generality, we write

$$h_i(\mathbf{s}) = h_i(s_i), \forall i \in [n]. \tag{32}$$

Since $p(\mathbf{s}|\mathbf{u})$ is a multivariate Gaussian, let $p(\mathbf{s}|\mathbf{u}) = \prod_{i=1}^{n} \frac{1}{Z_i^p} \exp\left(-\theta_{i,1}^p(\mathbf{u})s_i - \theta_{i,2}^p(\mathbf{u})s_i^2\right)$. Substitute this expression and the equation above into Eq. 17, we have

$$\theta_{i,1}^p(\mathbf{u})s_i + \theta_{i,2}^p(\mathbf{u})s_i^2 = \theta_{i,1}^q(\mathbf{u})h_i(s_i) + \theta_{i,2}^q(\mathbf{u})h_i(s_i)^2 + C_i, \forall i \in [n], \tag{33}$$

where $C_i$ is a constant. This obviously leads to

$$h_i(s_i) = a_i s_i + b_i, \forall i \in [n]. \tag{34}$$

To conclude, $\mathbf{g} \circ \mathbf{f}$ is a composition of a point-wise linear transformation and a permutation. □

## B  A CLASS OF IDENTIFIABLE NVP MIXING FUNCTIONS

Even if the mixing function is non-volume-preserving (NVP), it still leads to identifiability when its Jacobian has the following form:

$$|\det \boldsymbol{J}_{\mathbf{f}}(\mathbf{s})| = \prod_{i=1}^{n} |h_i(\mathbf{s}_i)| > 0, \tag{35}$$

where $h_i$ is arbitrary function satisfying $|h_i(\mathbf{s}_i)| > 0$. Obviously this class of function is a extension of volume-preserving transformations. To see this, note that when $|h_i(\mathbf{s}_i)| = 1$, this class of function degenerate into volume-preserving transformations.

Substitute the equation above into the change of variable rule $p(\mathbf{s}|\mathbf{u}) = q(\mathbf{f}(\mathbf{s})|\mathbf{u})|\det \boldsymbol{J}_{\mathbf{f}}(\mathbf{s})|$, and take the logarithm, we have

$$\sum_{i=1}^{n} \left( \log p_i(\mathbf{s}_i|\mathbf{u}) + \log |h_i(\mathbf{s}_i)| \right) = -\sum_{i=1}^{n} \left( \theta_{i,1}^{q}(\mathbf{u}) f_i(\mathbf{s}) + \theta_{i,2}^{q}(\mathbf{u}) f_i(\mathbf{s})^2 + \log Z_i^{q} \right). \tag{17'}$$

Using this equation, the proof of Lemma 1 still holds, because the in the proof we only use the fact that the left hand side of Eq. 17 has no cross terms, which is also the property of the equation above.

Therefore, the class of mixing functions defined above also leads to identifiability, even if it is non-volume-preserving. This means that there exist some non-trival extensions of our theory.

## C  SIMPLIFIED LOSS FUNCTION

Consider when $\mu_i(\mathbf{u})$ and $\sigma_i(\mathbf{u})$ are trainable, their optimal solutions of Eq. 9 have the closed forms as follows:

$$\begin{aligned} \mu_i^*(\boldsymbol{u}^{(k)}) &= \mathbb{E}_{(\mathbf{x},\boldsymbol{u}^{(k)})\in\mathcal{D}} \left[ g_i(\mathbf{x};\boldsymbol{\theta}) \right], \\ \sigma_i^*(\boldsymbol{u}^{(k)}) &= \sqrt{\mathbb{E}_{(\mathbf{x},\boldsymbol{u}^{(k)})\in\mathcal{D}} \left[ (g_i(\mathbf{x};\boldsymbol{\theta}) - \mu_i^*(\boldsymbol{u}^{(k)}))^2 \right]}, \end{aligned} \tag{36}$$

where $k = 1, 2$. Note that the optimal solutions depend on the parameters $\boldsymbol{\theta}$, hence when given $\boldsymbol{\theta}$, we denote $\mu_i^*(\boldsymbol{u}^{(k)})$ and $\sigma_i^*(\boldsymbol{u}^{(k)})$ by $\mu_i(\boldsymbol{u}^{(k)}, \boldsymbol{\theta})$ and $\sigma_i(\boldsymbol{u}^{(k)}; \boldsymbol{\theta})$ respectively. Substitute these optimal solutions to the loss function above, we finally obtain:

$$\mathcal{L}(\boldsymbol{\theta}) = \sum_{k=1}^{2} \sum_{i=1}^{n} \log \sigma_i(\boldsymbol{u}^{(k)}; \boldsymbol{\theta}). \tag{37}$$

In experiments, we only sample a mini-batch $\mathcal{B}$ from the full dataset $\mathcal{D}$ for each iteration, hence in this case $\sigma_i(\boldsymbol{u}^{(k)}; \boldsymbol{\theta}) = \sqrt{\mathbb{E}_{(\mathbf{x},\boldsymbol{u}^{(k)})\in\mathcal{B}} \left[ g_i(\mathbf{x};\boldsymbol{\theta}) - \mu_i(\boldsymbol{u}^{(k)};\boldsymbol{\theta}) \right]^2}$, which is the biased standard deviation of $\{g_i(\mathbf{x};\boldsymbol{\theta})\}_{(\mathbf{x},\boldsymbol{u}^{(k)})\in\mathcal{B}}$.

### C.1  DIMENSION REDUCTION

Here we will demonstrate that the simplified loss above leads to dimension reduction. This is a vital property for identifiability. On the one side, dimension reduction enable the learning algorithm to estimate the unknown dimensionality of data manifold. On the other, if we use a flow-based model to identify the true sources, then dimension reduction of latent variables $\mathbf{z}$ is necessary, as in flow-based models the dimensionality of $\mathbf{z}$ ($d$) is far more than the number of true sources ($n$).

In a flow-based model, the simplified loss is

$$\mathcal{L}(\boldsymbol{\theta}) = \sum_{k=1}^{2} \sum_{i=1}^{d} \log \sigma_i(\boldsymbol{u}^{(k)}; \boldsymbol{\theta}) = \sum_{k=1}^{2} \log \prod_{i=1}^{d} \sigma_i(\boldsymbol{u}^{(k)}; \boldsymbol{\theta}). \tag{38}$$

Hence minimizing the simplified loss is equivalent to minimizing $\prod_{i=1}^{d} \sigma_i(\boldsymbol{u}^{(k)}; \boldsymbol{\theta})$. To minimizing this term, most latent variables should have almost zero standard deviations.

Note that in implementation, the standard deviations of latent variables could not be exact zero, and not all standard deviations can be almost zero. First, in flow-based models, the data points are usually augmented by adding small noise, and hence the dimensionality of augmented data manifold is exactly $d$. As a result, the dimensionality of the corresponding latent space is also $d$, hence exact zero standard deviation is not possible due to the reversibility of flow-based models. Moreover, as we assume the transformation $\mathbf{g}$ is volume-preserving, the entropy of data distribution and latent distribution should be equal, and hence there must exist some latent variables with high standard deviations.

Therefore, the simplified loss will assign almost zero standard deviations to most latent variables while preserving some high standard deviations. A natural solution is to assign high standard deviations to meaningful latent variables, and assign almost zero standard deviations to redundant dimensions. This is exactly the dimension reduction ability of the simplified loss.

### C.2 INDEPENDENCE ENHANCEMENT

Next we will prove that the simplified loss can enhance the independence of different components of $\mathbf{z}$. This is a desirable property, as $p(\mathbf{z}|\mathbf{u})$ should be factorial according to our theory.

Our proof is based on Hadamard's inequality (Cover & Gamal, 1983): for any $n \times n$ matrix $\boldsymbol{K}$, $\det \boldsymbol{K} \leq \prod_i K_{i,i}$ with equality iff $K_{i,j} = 0$, $\forall i \neq j$. Let $\boldsymbol{K} = \mathbf{Cov}(\mathbf{z}, \mathbf{z})$, then we have $\det \mathbf{Cov}(\mathbf{z}, \mathbf{z}) \leq \prod_i \mathbf{Var}(\mathbf{z}_i)$. Then the simplified loss has a lower bound as follows:

$$\mathcal{L}(\boldsymbol{\theta}) = \frac{1}{2} \sum_{k=1}^{2} \sum_{i=1}^{n} \log \mathbf{Var}_{\boldsymbol{u}^{(k)}}(\mathbf{z}_i) \geq \frac{1}{2} \sum_{k=1}^{2} \log \det \mathbf{Cov}_{\boldsymbol{u}^{(k)}}(\mathbf{z}). \tag{39}$$

where the subscripts $\boldsymbol{u}^{(k)}$ represent the conditions.

When the simplified loss is optimized, the gap between it and its lower bound will become zero, and hence the equality holds. This means $\mathbf{Cov}(\mathbf{z_i}, \mathbf{z_j}) = 0$, $\forall i \neq j$. Therefore, minimizing the simplified loss can enhance the independence of different latent variables by reducing correlations.

In addition, note that our derivation above does not depend on the Gaussianity of $q(\mathbf{z}|\mathbf{u})$, and hence our conclusion is general.

## D  DETAILS ABOUT THE ARTIFICIAL AND SYNTHESIZED DATASETS

Table 3: Means and variances of each class in Triangles. $\alpha$ and $\beta$ represents uniform random variables on $[-1, 1]$ and $[1, 2]$, respectively.

| Factors | Value Ranges | Means | Variances |
|---|---|---|---|
| Rotation | $[-\pi, \pi]$ | $0 + 0.01\pi\alpha$ | $0.03\pi\beta$ |
| Width | $[1, 32]$ | $18 + 2\alpha$ | $1\beta$ |
| Height | $[1, 32]$ | $18 + 2\alpha$ | $1\beta$ |
| Gray Level | $[0, 255]$ | $170 + 5\alpha$ | $21\beta$ |

To generate the the artificial data, following (Sorrenson et al., 2020), we synthesize the sources and observed data by two steps: i) First, two sources are sampled from a 2-dim mixture of two Gaussians with their mean both $(0.0, 0.0)$, and with variances $(1.0, 0.5)$ and $(0.5, 1.0)$, respectively. From each Gaussian, $5,000$ points are sampled. Then an 8-dim standard Gaussian noise scaled by $0.01$ is concatenated with them. ii) The observed data is generated from the 10-dim sources, by a randomly initialized GIN (Sorrenson et al., 2020) with 8 fully connected coupling blocks.

To quantitatively test our method on image datasets, we have synthesized a gray-scale image dataset called "Triangles". Specifically, we choose to generate a grey-scale $32 \times 32$ image dataset of 2-D shapes similar with dSprites (Matthey et al., 2017) as shown in Fig. 6, in which all shapes are right triangles generated from 4 factors: rotation, width (length of the horizontal edge), height (length of the vertical edge) and gray level. These factors are viewed as sources, and the observations of sources are sampled from a mixture of two Gaussians with random means and variances. The value ranges of four factors are set as $[-\pi, \pi]$, $[1, 32]$, $[1, 32]$ and $[0, 255]$, respectively.

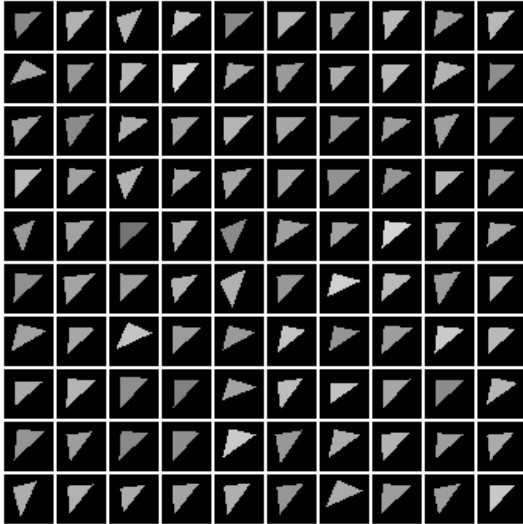

Figure 6: Samples from Triangles.

According to our requirement, in each class these factors follow a factorial multivariate Gaussian. To satisfied condition (iii) in Theorem 1, we set the variance of each factor as a proper random value, and so do the mean because we found in experiments that the condition (ii) in Theorem 1 is not necessary. The particular parameters of the four factors are shown in Table 3

The generating process of each image consists of three steps: i) for each class, randomly select four means and four variances according to Table 3, and obtain a 4-dim Gaussian; ii) sample one point from the 4-dim Gaussian, which corresponds to a right triangle; iii) judge what pixels locate inside the triangle according to the given rotation, width and height, and then assign them with the given gray level.

## E  QUALITATIVE RESULTS OF ABLATION STUDY

Here we show the qualitative results of our ablation study (see Fig. 7) to support our discussion in an intuitive way. In these figures, reconstruction is the estimated latent variables with top two standard deviations, as Sorrenson et al. (2020) reported that estimated latent variables with higher standard deviations are probably more meaningful. Spectrum is the sorted standard deviations of latent variables. The spectrum of the estimated latent variables is in black, while the spectrum of true latent variables is in grey.

## F  QUALITATIVE RESULTS ON CELEBA

To further show the performance of our framework on real-world datasets, we conduct experiments on CelebA (Liu et al., 2015) using GIN. Since our framework requires two distinct classes, we use all images in CelebA as one class, and their mirror images as another class. Such setting ensures the two obtained classes are distinct and have non-zero overlap. Similar with experiments on MNIST, we pick latent dimensions with high standard deviations as estimated latent variables, and plot them to show their corresponding underlying factors.

As shown in Fig. 8, the estimated latent variables with top 4 standard deviations are highly interpretable, corresponding to lighting on the left, lighting on the right, skin color, background brightness, hair (bang) and grin, respectively. These results further demonstrate that our framework is probably able to identify the true sources from real-world datasets, and hence support our theory.

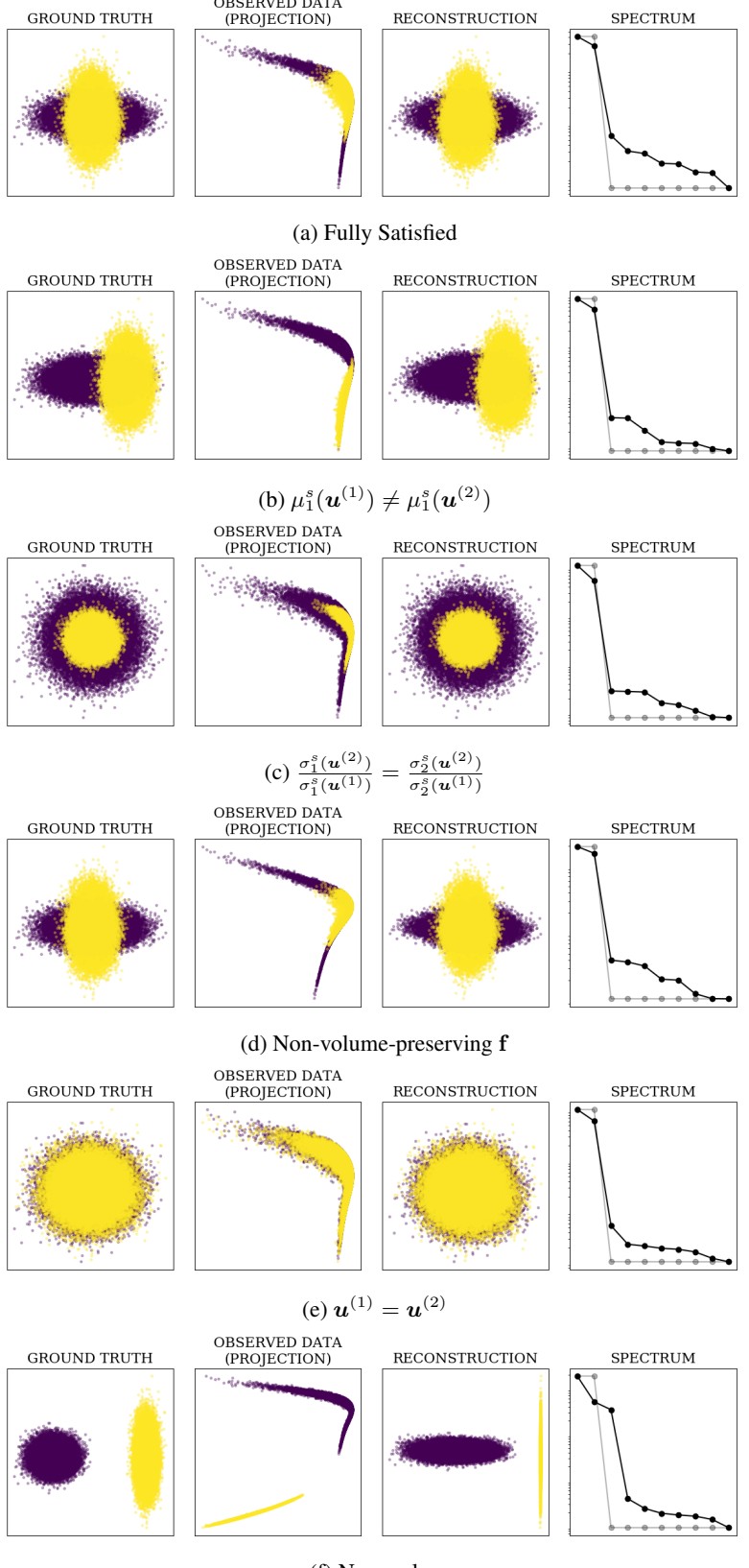

Figure 7: Qualitative results of ablation study.

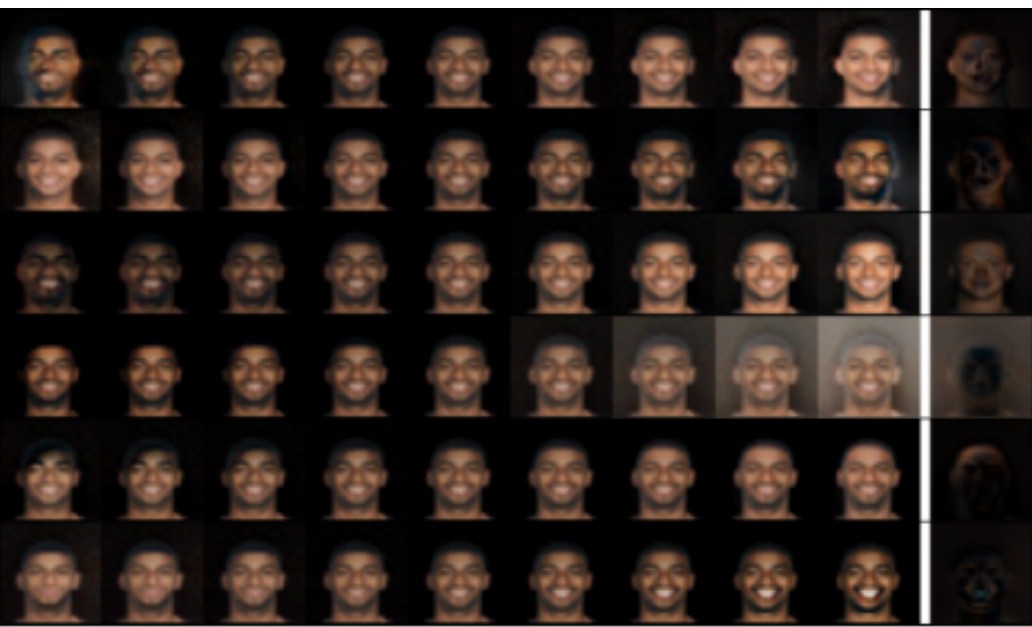

Figure 8: Traversals of estimated latent variables with top 6 standard deviations by GIN on CelebA. Each row is traversal by manipulating one estimated latent variable. The six rows represent lighting on the left, lighting on the right, skin color, background brightness, hair (bang) and grin, respectively.

