# OpenReview forum: "Nonlinear ICA Using Volume-Preserving Transformations"
_ICLR.cc/2022/Conference — ICLR 2022 Poster_

### Official Review · Reviewer_7dbX · 2021-10-24

**Correctness:** 4
**Technical Novelty And Significance:** 3
**Empirical Novelty And Significance:** 3
**Recommendation:** 6
**Confidence:** 3

**Main Review:**

I will admit that I'm not familiar with the recent developments when it comes to ICA and to Flow-based modeling of densities, but this seems like a good paper. It is difficult for me to assess the importance of the original contribution from the authors.

Q: Does the term "point-wise linear" refer to the kind of model described in https://arxiv.org/pdf/2001.06988.pdf ?

**Summary Of The Paper:**

The authors reformulate ICA in a way that makes it approachable with Flow-based methods. This yields a practical approach that the authors can apply to MNIST and to another dataset of that level of simplicity.

**Summary Of The Review:**

This is a good paper even though I do not have familiarity with the recent developments on that specific topic.

---

> ### Author Response · Authors · 2021-11-16
> **Response to Reviewer 7dbX**
>
> Thank you for the time and the nice review. We are encouraged that you appreciated our work. We hope that our response can address your questions. Please let us know if there are any further questions.
>
> Q1: What does 'point-wise' mean?\
> A1: In our work, 'point-wise' means point-to-point (dimension-to-dimension) mapping without cross terms and permutations. For example, if $(z_1,z_2)=(as_1+b, cs_2+d)$, then we say there exist a point-wise transformation between $(z_1,z_2)$ and $(s_1,s_2)$. In the literature, such property is also called 'component-wise'.

---

### Official Review · Reviewer_xbzu · 2021-10-29

**Correctness:** 3
**Technical Novelty And Significance:** 2
**Empirical Novelty And Significance:** 2
**Recommendation:** 6
**Confidence:** 4

**Main Review:**

I think there are some merits in terms of the novelty of the paper. However, the setting for ICA appears to be different from the classical setup as it requires labels. In the classical ICA setup, there appears to be no labels in the dataset. Understandably you are trying to solve the same problem but you have labels to train the model.

My questions are:
(1) How do you select $n$ in Equation (8)

(2) How do you obtain \mu (u) and \sigma (u) in equation (8)

(3) So does this approach still work if we don't have any labels for the data?

(4) Since the transformations are volume preserving, does that mean you recover the source signals in the correct scale?

**Summary Of The Paper:**

The authors propose to use volume preserving transformation to solve the disentanglement problem in latent variable models such as ICA. The authors have explained that in literature that ICA can be expressed as a factorial member of the exponential family, however, this arrives at a solution which is not identifiable. The authors have explained that there are two ways of achieving disentanglement, this could be done by either (1) restricting the sources or (2) restricting the function f.


Interestingly that authors have proposed to use a volume-preserving transformation typically found in normalizing flows to be able obtain identifiability. There is a volume preserving transformation from the sources to the mixed signal which represents the generative process. Then there is a volume preserving transformation from the mixed signal to the latent space (which is not the source signal). Then a factorial multivariaate Gaussian is used to identify the conditional independence of each source signal.

In the classical ICA problem setup, the dataset does not contain any labels. While in this setting the input dataset for ICA appears to contain labels (u?).

**Summary Of The Review:**

The paper proposes a simple and elegant solution to the disentanglement problem encountered in ICA. So I recommend in accepting.

---

> ### Author Response · Authors · 2021-11-16
> **Response to Reviewer xbzu**
>
> Thank you for the time and the valuable comments. We are pleased that you found our work interesting and elegant. In the following response, we answer your comments/questions point-by-point. Please let us know if there are any further questions.
>
> Q1: The setting for ICA is different from the classical setup, as labels are required in our work.\
> A1: We follow the setting proposed by Hyvarinen et al. [1], which has been a useful setting for nonlinear ICA theory.
>
> Q2: How to select $n$ in Eq. 8?\
> A2: $n$ is the number of sources (also known as the intrinsic dimension of data manifold), which is unknown in the estimating model of our framework. Actually, the selection of $n$ is unnecessary when estimating latent variables, as the algorithm can reduce the dimension of the estimated latent variables. This is one of the interesting properties of the simplified loss function (Eq. 10), which can naturally assign almost zero standard deviations to redundant dimensions. As a result, the number of non-zero standard deviations will approximate $n$, and the corresponding dimensions represent the estimated latent variables. In addition, if we exactly know how large $n$ is, it is possible to design a more powerful algorithm to identify the true sources.
>
> Q3: How to obtain $\mathbf{\mu}(\mathbf{u})$ and $\mathbf{\sigma}(\mathbf{u})$ in Eq. 8?\
> A3: We set these parameters to be trainable, and obtain their optimal solutions in Eq. 36. As shown in Eq. 36, for each batch of samples, the optimal $\mathbf{\mu}(\mathbf{u})$ and $\mathbf{\sigma}(\mathbf{u})$ are the average and the standard deviation in latent space, respectively. Therefore, in each iteration of the learning algorithm, $\mathbf{\mu}(\mathbf{u})$ and $\mathbf{\sigma}(\mathbf{u})$ are simply the average and the standard deviation of the learned representations, respectively.
>
> Q4: Does our approach still work if we don't have any labels for the data?\
> A4: Our theory requires two distinct classes of data and their labels to resolve the indeterminacy of rotation of the estimated latent variables (see Fig. 2). If we don't have any labels (i.e. there only exists one class of data), then theoretically we can easily construct an unidentifiable example. Interestingly, we empirically found that for some datasets (include the artificial dataset and MNIST), even though only one class of data is given, the true sources can be successfully identified. This implies that there exist some implicit biases to reduce the indeterminacy of rotation even when only one class of data is given. One bias might be the mini-batch sampling, and we leave this conjecture for future works. However, for our synthesized dataset Triangle, when given only one class, the identifiability of true sources will sharply decline both qualitatively and quantitatively compared with the case of two classes. This indicates that implicit biases cannot fully replace the supervised information of labels.
>
> Q5: Since the transformations are volume-preserving, are the source identified in the correct scale?\
> A5: The answer is NO. We say the sources $\mathbf{s}$ can be identified up to nonlinear/linear transformations, which means the estimated latent variables have the indeterminacy of nonlinear/linear transformations, including scaling. In other words, the scales of the estimated latent variables are non-deterministic. For example, if $(z_1, z_2)$ successfully identify the true sources, then so do $(z'_1= 0.5z_1, z'_2= 2z_2)$, and $(z'_1, z'_2)$ is the volume-preserving transformation of $(z_1, z_2)$. In addition, volume-preserving is a weak condition as it only reduce $1$ free degree of the Jacobian of function, and hence it is impossible to restrict the scales of all dimensions.
>
> [1] Aapo Hyvarinen, Hiroaki Sasaki, and Richard Turner. Nonlinear ica using auxiliary variables and generalized contrastive learning. In The 22nd International Conference on Artificial Intelligence and Statistics, pp. 859–868. PMLR, 2019.

---

### Official Review · Reviewer_kNre · 2021-10-31

**Correctness:** 3
**Technical Novelty And Significance:** 3
**Empirical Novelty And Significance:** 3
**Recommendation:** 6
**Confidence:** 3

**Main Review:**

The nonlinear ICA problem is of significant research interest, and it is important to provide an identifiability guarantee for nonlinear ICA. In this work, the authors impose some reasonable assumptions on both the sources and mixing functions and provide two theorems for nonlinear and linear identifiability respectively. In particular, the theoretical results lead to the insight that the main indeterminacy of a nonlinear ICA framework using volume-preserving transformations is the rotation of latent variables. The authors further perform experiments to show that their method outperforms the state-of-the-art nonlinear ICA method iVAE.

This paper is well-written, and the motivations and contributions are clearly presented.

**Summary Of The Paper:**

In this work, the authors propose a framework for nonlinear ICA, in which the mixing function is a volume-preserving transformation, and the conditions for the sources can be relaxed compared to some prior works. The authors prove the identifiability of the proposed framework and implement the framework by volume-preserving Flow-based models. Numerical experiments on both synthetic and real data are performed to corroborate the theoretical results.

**Summary Of The Review:**

Overall, my impression is that this work provides some solid contributions to the interesting nonlinear ICA problem. One issue I hope the authors can address is including the source codes in a code appendix.

---

> ### Author Response · Authors · 2021-11-16
> **Response to Reviewer kNre**
>
> Thank you for the time and the nice reviews. We are encouraged that you acknowledged the significance of our focused problem and our solid contributions. We hope that our response can address your questions. Please let us know if there are any further questions.
>
> Q1: Include the source codes in a code appendix.\
> A1: Thanks for your interest in our work. Our code is based on GIN [1], the main difference is we only need two classes of data to train the mixing model. We will release our code soon.
>
> [1] Peter Sorrenson, Carsten Rother, and Ullrich Kothe. Disentanglement by nonlinear ica with general incompressible-flow networks (gin). In International Conference on Learning Representations, 2020.

---

> > ### Comment · Reviewer_kNre · 2021-11-29
> > **Response**
> >
> > Thanks for the authors' response. Based on the response and other reviewers' comments, I would like to keep my score unchanged.

---

### Official Review · Reviewer_kWud · 2021-11-02

**Correctness:** 4
**Technical Novelty And Significance:** 3
**Empirical Novelty And Significance:** 2
**Recommendation:** 6
**Confidence:** 3

**Main Review:**

Strengths:
-	The authors provide theoretical support for the use of the pre-existing GIN model for ICA. They propose novel assumptions and prove this leads to identifiability.
-	The paper is very clearly written.

Weaknesses:
-	The authors claim that a volume-preserving mixing function is a natural restriction and is easily satisfied. I would like to see a stronger argument why this is true, as it seems easy to think of non-volume-preserving mixing functions. Such an argument should include why the triangle dataset and MNIST would be generated by volume-preserving mixing functions.
-	The experiments find that none of the assumptions in the main theorem are necessary for identifiability to hold. More discussion about what necessary conditions would look like would improve the paper.
-	The experiments are not very strong. Only quantitively analyzes the simple triangle dataset, so does not include a non-trivial dataset with known sources, so that identifiability can be quantified. Also, the experiments show large overlap with the experiments of [Sorrenson et al 2020]. An additional experiment with more complicated images with known sources would improve the paper.
-	It is unclear why the model does not fully succeed in identifying the true sources in the triangle dataset. Is one of the assumptions not satisfied? Are there learning difficulties?

Further comments:
-	It seems that there is a constraint that q(z|u) must be equal to the push-forward of p(s|u) through $g \circ f$, in other words, that Figure 1 can be interpreted as a commuting diagram. However, this is never explicitly stated in the definition of the estimating model. Could the authors please clarify this?
-	I believe the appendix should be separately provides as supplementary material.

Typos:
-	In (18) in the appendix, z_0 should be s_0?
-	Above (23) in the appendix, should positive-defined be positive definite? Or positive semi-definite?


**Summary Of The Paper:**

[Sorrenson et al 2020] proposed to do nonlinear ICA via volume preserving mixing functions, when a variable u is observed that makes the latent variables conditionally independent. Via maximum likelihood optimization, the ICA model uses a volume-preserving encoder and then maximises a latent distribution, which is a factorized Gaussian conditional on the observed variable u.

The submitted paper does a theoretical identifiability analysis on that procedure. It assumes that the data is generated by an independent distribution, given observed u, then mixed by a volume-preserving mixing function. Furthermore, it assumes that both the mixer and the encoders are twice differentiable, there exist two observed u variables, u^1 and u^2 such that that the two resulting distributions of encoded latent variables have the same mean and that the set of ratios of the two standard deviations for each variable is distinct. If these assumptions hold, the paper proves that the original sources are identifiable up to a permutation and dimension-wise diffeomorphism.

The authors do three experiments: 2D mixture of Gaussians, triangle images and MNIST. For the first two, the authors quantitatively estimate the source identification and for all datasets, qualitative results on identification are given.

**Summary Of The Review:**

The paper provides an interesting theoretical analysis of an existing method. I weakly recommend acceptance, as the assumptions are not sufficiently motivated, the assumptions are shown to not be necessary and the experimental evaluation is not very strong.

---

> ### Author Response · Authors · 2021-11-16
> **Response to Reviewer kWud**
>
> Thank you for the time, through comments and valuable suggestions. We are encouraged that you acknowledged the novelty and theoretical soundness of our work. In the following response, we answer your comments/questions point-by-point. Please let us know if there are any further questions.
>
> Q1: Why volume-preserving mixing function is a natural restriction?\
> A1: First of all, it was empirically reported that volume-preserving flows (VPFs) can generate digit images and faces images [1], and hence volume-preservation is a rather loose restriction. More importantly, from the perspective of degree of freedom, the restriction of volume-preservation only reduces $1$ free degree of the Jacobian of mixing function, which is negligible compared with the whole free degree ($n\times n$). Therefore, volume-preservation is not a rigid restriction both empirically and theoretically. Furthermore, in our theory, the volume-preservation of mixing function $\mathbf{f}$ is not harmful to the generative ability of generative models. To see this, note that in Theorem 1, the priors of sources have no particular forms, and almost any datasets can be generated by volume-preserving mixing functions from a complicated enough prior. Furthermore, in Appendix B we provide a much more general condition for mixing function.
>
> Q2: Why the Triangle dataset and MNIST would be generated by volume-preserving mixing functions, and why the model does not fully succeed in identifying the true sources in Triangle.\
> A2: If the priors of sources are fixed, there may not exist a volume-preserving mixing function to generate the given dataset. On the other hand, if we do not restrict the priors, it is flexible to choose the mixing function to be volume-preserving . In our artificial dataset, the data points are generated by a volume-preserving flow, and hence the sources are perfectly identifiable. For our synthesized Triangle dataset, the priors of sources are set as Gaussians, thus the generative process is hard to be volume-preserving. We think this is the main reason why the MCC by GIN on this dataset is not perfectly $1$. Specifically, MCC is used to evaluate the linear correlation between $\mathbf{s}$ and $\mathbf{z}$, while the linear correlation depends on both the Gaussian priors and a restriction on mixing function (volume-preservation or the condition in Appendix B) according to Theorem 2. As for MNIST, it can be well generated by a volume-preserving flow from Gaussian priors [1], and hence we can assume its priors are Gaussians and its mixing function is volume-preserving.
>
> Q3: The necessity of our proposed conditions.\
> A3: Please refer to the answer A3 in response to Reviewer eqBB.
>
> Q4: An additional experiment with more complicated images with known sources.\
> A4: The main goal of this work is to explore a new direction for nonlinear ICA, and provide some insights for this topic, so we tend to leave experiments with more complicated datasets to future works. On the other side, most of the complicated datasets do not satisfy the essential conditions in our theory: two classes and non-zero overlap. Therefore, in these complicated datasets, it is difficult to identify the true sources without substantial improvement in the learning algorithm. Such improvement, in our opinion, will be a vital step towards the application of nonlinear ICA and will be left to future works.
>
> Q5: A constraint that $q(\mathbf{z}|\mathbf{u})$ must be equal to the push-forward of $p(\mathbf{s}|\mathbf{u})$ through $\mathbf{f}\circ\mathbf{g}$.\
> A5: This is true, and we have mentioned this point in the last paragraph of Section 3, i.e. the form of $q(\mathbf{z}|\mathbf{u})$ implicitly restrict the form of $p(\mathbf{s}|\mathbf{u})$ through $\mathbf{f}\circ\mathbf{g}$. For example, if $\mathbf{f}\circ\mathbf{g}$ is volume-preserving, then the support sets of $q(\mathbf{z}|\mathbf{u})$ and $p(\mathbf{s}|\mathbf{u})$ should have the same volume. However, note that if the restriction on mixing function $\mathbf{f}$ is reduced to the condition in Appendix B, then the constraint on $q(\mathbf{z}|\mathbf{u})$ does not restrict the form of $p(\mathbf{s}|\mathbf{u})$.
>
> Q6: The appendix should be separately provided as supplementary material.\
> A6: Thanks for pointing out this. We will correct it in the revised version.
>
> [1] Laurent Dinh, David Krueger, and Yoshua Bengio. Nice: Non-linear independent components estimation. arXiv preprint arXiv:1410.8516, 2014.

---

> > ### Comment · Reviewer_kWud · 2021-11-19
> > **Shouldn't identifiability be about finding the true priors and decoders of real-world data? Do your assumptions apply to those?**
> >
> > Thank you for your response.
> >
> > Re Sorrenson: thank you for highlighting the differences. Perhaps you can elaborate more on how you improve on that work in your paper.
> >
> > Re A1 & A2: The fact that volume-preserving flows can model rich data distributions - esp with a complicated prior, seems a bit besides the point to me. I think the relevant question of identifiability is: of the data generating processes in the real world we would like to model, where we separate a prior distribution and a decoder, can we recover the prior and decoder from the data. We can do that with your method only if the *real* decoder is volume-preserving. And I wonder how often that is true. Your image task seems a good example where this is false. First, you posited some Gaussian prior then use a rendering method to generate the images. The question then is: from data, can we recover that Gaussian prior and the rendering method? It's far from obvious to me why that would be true. Maybe the fact that your don't perfectly solve the Triangles task indicates that your assumptions don't apply to that task?
> > The same questions can be asked about the three assumptions in theorem 1: you motivate why they are sufficient to make your theorem work, but you don't motivate why any real data generating process satisfies them.
> > The paper could be greatly improved if you made it clear why real world data generating processes satisfy your assumptions.
> >
> > Re A5: I meant that the exposition in the paper of this fact, that q must be the push-foward of p, can be clarified. In the paper, you mention a consequence of this fact, not the fact itself.
> >
> > Additional question: when you say in eqn (4) that f is volume-preserving, which metric tensor do you assign to manifold M? Is it the metric induced by the embedding in R^d?
> >
> > I am not sufficiently convinced about the assumptions and keep my score.

---

> > > ### Author Response · Authors · 2021-11-20
> > > **The generating process of a real-world dataset can be arbitrary when both the forms of true prior and decoder are unknown.**
> > >
> > > Thanks for your thorough comments and valuable suggestions. In the following response, we answer your comments/questions point-by-point. Please let us know if there are any further questions.
> > >
> > > Q1: Shouldn't identifiability be about finding the true priors and decoders of real-world data?\
> > > A1: In nonlinear ICA, when merely given a dataset, it is impossible to identify both the true prior and the true decoder without giving the particular forms of the prior or decoder. The key point here is that given a dataset with $\textbf{unknown}$ generative process, there exist infinite pairs of prior and decoder to generate it. To see this, assume we have obtained a possible pair denoted by $(p(\mathbf{s}),\mathbf{f})$, then we can construct another possible pair $(p'(\mathbf{s}'),\mathbf{t}\circ\mathbf{f})$, where $\mathbf{t}$ is an arbitrary transformation (satisfied the constraints on decoder), $\mathbf{s}'=\mathbf{t}^{-1}(\mathbf{s})$ and $p'$ is its density function. Therefore, when talking about identifying the true prior and the true decoder for a given dataset, the true decoder and the true prior should be given, respectively. As a result, for a given dataset, if we ask whether its true decoder (mixing function) is volume-preserving, we should first fix its true prior. Otherwise, if the prior is arbitrary, then any volume-preserving transformations are possible decoders. Note that in Theorem 1 in our paper, the prior $p(\mathbf{s})$ is arbitrary, and hence the constraint on the mixing function (volume-preservation or the condition introduced by Appendix B) is suitable for any dataset.\
> > > On the other side, given a dataset, if we assume the true prior is Gaussian as introduced in Theorem 2, then the true decoder (mixing function) might not be volume-preserving (or satisfy the condition introduced by Appendix B). This also explains why the MCC on Triangles by GIN is not perfectly $1$, as MCC mainly evaluates the linear correlation between $\mathbf{z}$ and $\mathbf{s}$, while in Theorem 2 the linearity depends on both the Gaussian prior and volume-preserving mixing function. In other words, if we fix the true prior of Triangles as Gaussian, the true decoder of Triangles might not be volume-preserving, which does not satisfy the condition in Theorem 2. However, this is not contradictory to our main theoretical result, as in Theorem 1 there is no constraint on the particular form of prior.\
> > > Overall, our main theoretical result (Theorem 1) is very general and suitable for most datasets, as most datasets can be generated by volume-preserving transformations (or the non-volume-preserving transformations introduced by Appendix B) from complicated priors. Our Theorem 2 is a special case of Theorem 1, which further assumes the true prior is Gaussian, and hence it is suitable for those datasets which are generated by volume-preserving transformations from Gaussian priors. For real-world datasets, most of them satisfy the conditions on the prior and mixing function in Theorem 1, but it is difficult to check whether they satisfy the conditions in Theorem 2.
> > >
> > > Q2: $q$ must be the push-forward of $p$.\
> > > A2: Thanks for pointing out this, and we will clarify this point in our revised version.
> > >
> > > Q3: Which metric tensor do you assign to manifold $\mathcal{M}$? \
> > > A3: The metric on manifold $\mathcal{M}$ is the metric induced by the embedding in ambient space $\mathbb{R}^d$.

---

> > > > ### Comment · Reviewer_kWud · 2021-11-29
> > > > **Thanks for response, my doubts remain**
> > > >
> > > > Thanks for your response. I am still not convinced of the set-up, especially in the image cases. Your actual generating process does not satisfy your assumptions, so you fail to identify the true prior and mixing function. Hence, I think your volume-preserving assumption actually is not suitable for many kinds of datasets.
> > > > Still, I think the theoretical contribution is interesting, so I weakly recommend acceptance.

---

> > > > > ### Author Response · Authors · 2021-11-30
> > > > > **Further clarification and summary**
> > > > >
> > > > > Thanks for your comments, and we would like to further clarify your concerns.
> > > > >
> > > > > Here we highlight that we have introduced a much weaker condition on mixing functions (see Appendix B). The introduced condition only requires the mixing functions to preserve the independence of factors, and hence is much more general than volume-preservation. Moreover, our Theorem 1 does not restrict the particular form of prior, hence it is suitable for most datasets. The identifiability guarantee of Theorem 1 is the core theoretical contribution of this paper, which has been approved by reviewers.
> > > > >
> > > > > As for Theorem 2, it is a special case of Theorem 1, which further restricts the prior to be a factorial Gaussian. We admit that some datatsets (including the synthesized image dataset `Triangles') do not satisfy the conditions of Theorem 2, hence the MCC on Triangles by our framework is not perfectly $1$. However, we believe that there might exist many different priors to ensure different identifiability, and are suitable for different datasets. We show in experiments that our framework can successfully identify the true sources of artificial data and MNIST, and achieve great success on Triangles even though this dataset does not satisfy some conditions of Theorem 2. This is our experimental contribution, and we believe that it is sufficient to show the applicability and power of Theorem 1.

---

### Official Review · Reviewer_eqBB · 2021-11-02

**Correctness:** 3
**Technical Novelty And Significance:** 2
**Empirical Novelty And Significance:** 2
**Recommendation:** 6
**Confidence:** 3

**Main Review:**

 The framework is mainly reused from Sorrenson et al.'s work. The authors simplify the loss derived by Sorrenson et al.; however, they do not make clear why this is necessary or desirable. Their main contribution are two novel identifiability theorems, which state that under a few more conditions in addition to the volume preserving mixing function the true sources can be identified. Empirically, not all of the conditions seem to be necessary. Even when a non-volume-preserving map is used, the true sources of synthetically generated data can be reconstructed in their ablation study in section 6.2.1. This ablation study raises the question whether the formulation of this framework for nonlinear ICA is to rigid, i.e. model used is too restricted.

The method outperforms iVAE, which is an important baseline in the field.

As a minor command, I want to add that there is a typo in the first sentence of section 6.2: it should be "variables", not "variabls".

**Summary Of The Paper:**

The paper proposes a framework for nonlinear ICA with that restriction that the mixing function is a volume preserving transformation. The authors prove that given this restriction as well as a few other conditions the sources are identifiable. The model set up is very similar to the one used in Sorrenson et al.'s work "Disentanglement by nonlinear ica with general incompressible-flow networks (gin)": A real NVP model is made volume preserving by a slight modification of the flow map and this flow is used to learn the mixing function. The sources are assumed conditional independent given an auxiliary variable $u$, which are the labels of classes in their experiments. The mixing function is learned by maximizing the Gaussian likelihood of the reconstruction sources, which is also conditionally independent given $u$.

The authors demonstrate empirically that some of the conditions of their identifiability proof might not be necessary because the sources can still be reconstructed empirically although the condition is not satisfied. Hence, identifiability might even hold true for an even larger class of problems. They also compare their method to iVAE and show that their framework can identify the true sources of generated data much better than this baseline. Furthermore, they apply the framework to MNIST and can identify a few important source variables explaining most of the variance of the dataset.

**Summary Of The Review:**

I appreciate the two novel identifiability theorems. However, the ablation study  in the experiments section lets the conditions appear to restrictive. Furthermore, it is not clear what distinguishes this work from Sorrenson et al.'s article apart form the identifiability theorems. Therefore, I tend towards rejecting this article but I am open to enter a discussion with the authors and the other reviewers.

Edit:
Given the helpful clarification for the authors to distinguish their work from Sorrenson et al.'s article I will raise my score by one.

---

> ### Author Response · Authors · 2021-11-16
> **Response to Reviewer eqBB**
>
> Thanks for your time and valuable comments. We are pleased that you acknowledged the novelty and theoretical soundness of our work. We hope that our response can address your questions. Please let us know if there are any further questions.
>
> Q1: The framework is mainly reused from Sorrenson et al.'s work [1].\
> A1: Our learning algorithm is similar to Sorrenson et al.'s work, as we utilize their proposed module called GIN and loss function to identify the sources. However, there are some essential differences between our framework and theirs. First, our framework only needs $2$ classes of data for identifiability due to the advantages of our theory, while Sorrenson et al.'s work requires $nk+1$ distinct classes, which is far more restrictive than our condition. Moreover, in our framework, the volume-preservation of mixing function is necessary according to our theory, while in Sorrenson et al.'s work, it is merely an auxiliary constraint comparable with orthonormality. In addition, we emphasize the necessity of non-zero overlap of different classes in this work, which is somewhat neglected in previous works.
>
> Q2: The necessity of the simplification of Sorrenson et al.'s loss function [1] is not clear.\
> A2: We want to show the ability of dimension reduction of the simplified loss function, and we have discussed the simplification in Appendix C. Specifically, the standard deviations of most dimensions will be reduced to almost zero when optimizing the simplified loss function, which is analog to PCA. We will provide a detailed discussion about the dimension reduction property in our revised paper.
>
> Q3: Whether the formulation of this framework for nonlinear ICA is too rigid?\
> A3: We discuss the main conditions in our theory one by one. Firstly, we conjecture the concentric property (condition (ii) in Theorem 1) is unnecessary, while a weaker condition that non-zero overlap of the given two classes is necessary, which is empirically proven in our ablation study. Secondly. the condition (iii) in Theorem 1 is obviously necessary in our theory, as it resolves the indeterminacy of rotation of the estimated latent variables, otherwise, an unidentifiable case can be easily constructed (see Fig. 2). There are some biases in deep learning which can also reduce the indeterminacy of rotation of the estimated latent variables, and we conjecture one of which is mini-batch sampling, but this is not contradictory to the necessity of our proposed condition (iii) in our theory. Finally, we find that MCC cannot fully characterize the identifiability of the mixing model. Actually, an arbitrary non-volume-preserving $\mathbf{f}$ can lead to unidentifiability, even though the corresponding MCC is $1$. To see this, note the qualitative result of non-volume-preserving $\mathbf{f}$ has some visual distortion (see Fig.7(d)). Therefore, in our theory the restriction on the mixing function is necessary, and we have proposed a stronger version (volume-preservation) and a weaker version (see Appendix B) in our paper.
>
> [1] Peter Sorrenson, Carsten Rother, and Ullrich Kothe. Disentanglement by nonlinear ica with general incompressible-flow networks (gin). In International Conference on Learning Representations, 2020.

---

> > ### Comment · Reviewer_eqBB · 2021-11-19
> > **Reply to Paper3031 Authors**
> >
> > Thank you for the clarifications. I changed my score accordingly.

---

### Author Response · Authors · 2021-11-16
**General comments**

We appreciate the reviewers' time and valuable comments. Overall, the reviewers thought our work well motivated and written, and acknowledged our novelty, theoretical soundness and comprehensive evaluations. The major concerns lie in the relation with Sorrenson et al.'s work, and the discussion about ablation study. We clarify the major concerns in the following discussion.

$\textbf{The differences between our framework and Sorrenson et al.'s work:}$ Our framework has some key differences compared with Sorrenson et al.'s work [1]. 1) Their work requires $nk+1$ distinct labels (values of $\mathbf{u}$) as their work is based on the theory developed by Hyvarinen et al [2]. and Khemakhem et al. [3]. Moreover, they assume that the priors of sources are members of exponential family. While in our framework, only two labels are sufficient for identifiability due to the advantages of our proposed theory. We also do not restrict the density function of each source $p_i(s_i|\mathbf{u})$ to be any particular functional form in our main theoretical result (Theorem 1). We are the first to significantly reduce the class numbers to two and remove the particular form of sources to our best knowledge. Based on our theoretical guarantees, the true sources can be successfully identified with only two classes of data on an artificial dataset, a synthesized image dataset and MNIST. 2) Our work emphasizes the necessity of a volume-preserving estimating function $\mathbf{g}$ for identifiability according to our theory, while in Sorrenson et al.'s work, volume-preservation is merely an auxiliary constraint comparable with orthonormality in linear transformations. Our framework is essentially different from Sorrenson et al.'s work, and our theoretical results significantly stronger than theirs.

$\textbf{Discussion about the ablation study:}$ As for the ablation study, we aim at pointing out that 1) the concentric property (condition (ii) in Theorem 1 is not necessary, but non-zero overlap of two classes is necessary, which reveals that our theorems can be further extended to non-concentric cases; 2) there exist some biases in deep learning to reduce the indeterminacy of rotation of learned representations (see Fig. 2), and in this work, we conjecture that mini-batch sampling might be one of the biases. However, this does not mean that condition (iii) is unnecessary in theory, because without such condition we can easily construct an unidentifiable case (Fig. 2); 3) there exists a broader class of mixing functions to guarantee identifiability beyond volume-preserving transformations, as shown in Appendix B.

$\textbf{Discussion about non-volume-preserving mixing functions:}$ For non-volume-preserving mixing functions on the artificial dataset, the quantitative (see Table 1) and qualitative results (see Fig. 7(d)) are not fully consistent. This demonstrates that MCC cannot fully characterize the identifiability of the mixing model. Specifically, we find that on the artificial dataset, even though MCC is 1.0 in the case of non-volume-preserving $\mathbf{f}$, the recovery of sources can be somewhat unsuccessful. To see this, note that in Fig. 7(d) (the qualitative result of non-volume-preserving $\mathbf{f}$), the recovery of ground truth has some visual distortions even though its MMC score is 1. This demonstrates that with our involving restrictions on the sources, an arbitrary non-volume-preserving mixing function $\mathbf{f}$ might lead to unidentifiability problem, hence involving some restrictions to $\mathbf{f}$ (like volume-preservation or the condition introduced in Appendix B) is necessary.

[1] Peter Sorrenson, Carsten Rother, and Ullrich Kothe. Disentanglement by nonlinear ica with general incompressible-flow networks (gin). In International Conference on Learning Representations, 2020.\
[2] Aapo Hyvarinen, Hiroaki Sasaki, and Richard Turner. Nonlinear ica using auxiliary variables and generalized contrastive learning. In The 22nd International Conference on Artificial Intelligence and Statistics, pp. 859–868. PMLR, 2019.\
[3] Ilyes Khemakhem, Diederik Kingma, Ricardo Monti, and Aapo Hyvarinen. Variational autoencoders and nonlinear ica: A unifying framework. In International Conference on Artificial Intelligence and Statistics, pp. 2207–2217. PMLR, 2020

---

### Author Response · Authors · 2021-11-23
**Summary of the revision**

We sincerely thank all the reviewers for the valuable comments, which help to improve the quality of our work. We summarize the revision in the updated version as follows:\
$\bullet$ We revised Section 1 to emphasize our technical contributions\
$\bullet$ We added discussion on a broader class of mixing functions for identifiability guarantee  (see Section 3.1)\
$\bullet$ We added discussion on the different essences of mixing functions' and estimating functions' volume-preservation (see Section 3.2)\
$\bullet$ We added discussion on the implicit constraints on priors (see Section 3.2)\
$\bullet$ We revised the typeface of the discussion about theoretical insight (see Section 4) to italics to highlight its importance \
$\bullet$ We added discussion on the simplified loss function to highlight its ability of dimension reduction and independence enhancement (see Section 5 and Appendix C)\
$\bullet$ We move the details of the generative processes of artificial data and synthetic images to Appendix D \
$\bullet$ We revised the discussion of the ablation study to emphasize its consistency with our theory (see Section 6.2.1) \
$\bullet$ We added qualitative results on CelebA to further support the advantages of our framework (see Section 6.3 and Appendix F)

---

### Decision · Program_Chairs · 2022-01-20

**Decision:**

Accept (Poster)

**Comment:**

This paper proposes an identifiable nonlinear ICA model based on volume-preserving transformations. The overall approach is very similar to the GIN method published @ ICLR 2020. There is a weak consensus among the reviewers that this paper has some merit, although none pushed for acceptance. After reviewing the paper myself, I agree that the contributions here appear to be incremental, but the results do push this growing field of identifiable latent variable models forward.